# Validation of two novel human activity recognition models for typically developing children and children with Cerebral Palsy

**Marte Fossflaten Tørring**[1,2]*, **Aleksej Logacjov**[3], **Siri Merete Brændvik**[1,4],
**Astrid Ustad**[1], **Karin Roeleveld**[1], **Ellen Marie Bardal**[1,4]

1 Department of Neuromedicine and Movement Science, Faculty of Medicine and Health Sciences, Norwegian University of Science and Technology, NTNU, Trondheim, Norway, 2 Physiotherapy Unit, Trondheim Municipal, Trondheim, Norway, 3 Department of Computer Science, Faculty of Information Technology and Electrical Engineering, Norwegian University of Science and Technology, NTNU, Trondheim, Norway, 4 Clinic of Rehabilitation, St Olavs Hospital, Trondheim University Hospital, Trondheim, Norway

* marte.f.torring@ntnu.no

**Data Availability Statement:** All metadata and raw acceleration files with annotations are available from Dataverse (https://doi.org/10.18710/

## Abstract

Human Activity Recognition models have potential to contribute to valuable and detailed knowledge of habitual physical activity for typically developing children and children with Cerebral Palsy. The main objective of the present study was to develop and validate two Human Activity Recognition models. One trained on data from typically developing children (n = 63), the second also including data from children with Cerebral Palsy (n = 16), engaging in standardised activities and free play. Our data was collected using accelerometers and ground truth was established with video annotations. Additionally, we aimed to investigate the influence of window settings on model performance. Utilizing the Extreme gradient boost (XGBoost) classifier, twelve sub-models were created, with 1-,3- and 5-seconds windows, with and without overlap. Both Human Activity Recognition models demonstrated excellent predictive capabilities (>92%) for standardised activities for both typically developing and Cerebral Palsy. From all window sizes, the 1-second window performed best for all test groups. Accuracy was slightly lower (>75%) for the Cerebral Palsy test group performing free play activities. The impact of window size and overlap varied depending on activity. In summary both Human Activity Recognition models effectively predict standardised activities, surpassing prior models for typically developing and children with Cerebral Palsy. Notably, the model trained on combined typically developing children and Cerebral Palsy data performed exemplary across all test groups. Researchers should select window settings aligned with their specific research objectives.

## 1. Introduction

Accurate measures of physical activity (PA) are crucial for evidence-based evaluation and regulation of PA in children and adolescents, as well as for assessing the effectiveness of PA interventions. Acceleration-based sensors have gained popularity in recent years as a supplement,

EPCXCC). To replicate our experiment one would also need our code, which is published on GitHub (https://github.com/ntnu-ai-lab/harth-ml-experiments/tree/main).

**Funding:** We have received financial support in form of a post-doctorate and doctorate position at the Faculty of Medicine and Health Sciences, Norwegian University of Science and Technology (Norwegian name: Fakultetet for medisin og helsevitenskap, Norges tekniske-Naturvitenskapelige Universitet) and from Vestfold Hospital Trust (Norwegian name: Sykehuset i Vestfold).

**Competing interests:** The authors have declared that no competing interests exist.

and a more reliable alternative to self-reported data in both intervention and population studies [1, 2]. Accelerometers provide acceleration in 3D, which can be used to estimate frequency, intensity, mode and duration of activity and sedentary behaviour, offering valuable insights into natural behaviour in children [3–6].

Traditionally, accelerometer data have been analysed using brand specific software with a count-based approach providing information about the intensity of the performed activities [7, 8]. Even though several population- and age-specific cut-off values for the different intensities have been developed, the validity is questionable both in typically developing (TD) children and in children with disabilities [9, 10]. Cut-off values for children with cerebral palsy (CP) has for example shown to misclassify 30–40% of the performed intensities [11]. An alternative approach is to classify unique behavioural patterns by using Human Activity Recognition (HAR) models, and thereby identify and recognise specific activity types, such as sitting, jumping, or standing [7, 12]. HAR models have shown promising results in recognising activities with over 90% accuracy in free living conditions for various population groups [12–16]. In studies involving children the accuracy is inconsistent and varies between 62–95% [8, 15–19]. This inconsistency is probably depending on factors like window size, activity protocol and algorithm used.

Despite the advantages with HAR models, challenges include the need for specific training data, large data sets, definition of optimal accelerometer placements, window size considerations and feature extraction [13, 20–22]. The HAR models also identifies unique movement patterns which may differ between adults and children. The accuracy of existing models trained on healthy adults may therefore be questioned for accurate recognition of PA in children, especially those with disabilities and deviating movement patterns [23, 24]. CP is one of the most frequent disability among children [25] with the global prevalence of approximately 1.6 /1000 live births in high income countries [26], and thereby it is suitable to include this group for our endeavour. Cerebral Palsy (CP) is a collective term for various neurological impairments resulting from cerebral injuries before the age of two [27–29]. The CP condition may include a wide range of physical impairments like reduced walking speed and stride length [30], impaired balance, [31] and secondary musculoskeletal impairments like contractures and skeletal deformities [32], all which may give deviating movement patterns compared to TD children.

The previous studies validating HAR models for TD children are hard to compare, due to use of different classifiers, window sizes, accelerometer placements and activity protocols [16, 17]. Moreover, accuracies often drop significantly when transitioning from laboratory-based to free-living conditions [15, 19]. Only four publications have validated HAR models in children with CP, and they present the same issues regarding the model specifications [24, 33–35]. All these studies achieved good accuracy for sedentary behaviour (over 80%), but variable results for free-living activities, with some as low as 27% accuracy [34]. The protocols used in these studies focus on sedentary behaviour and walking paces and do not adequately represent free-living childlike activities, lacking elements like ballgames, outdoor activities, and free play [24, 33–35]. Therefore, a new HAR model validated on free living data is needed for TD children and one spesificaly for children with CP.

Various human activities unfold over different durations, which the HAR models need to detect. For instance, a single step takes about 500 milliseconds to complete [36]. However, walking as an activity involves multiple steps, and thereby extending the time frame beyond a mere 1-second interval. Consequently, the choice of window size is a contentious issue, impacting accuracy for both cut-off, deep-learning and HAR methods, particulary in children, who change activity frequently [22, 23, 37]. Some studies suggest increased accuracy with larger window sizes (e.g., 15 seconds), while others argue for shorter windows (1–2 seconds)

[15, 18]. Addtionally the use of overlapping windows, which provides more data points, is in some cases favoured for better classification [38, 39].

To meet the challenges described with HAR, the present study aimed to develop and validate two machine learning models for recognizing habitual activities in TD children and ambulatory children with CP. Specifically, the two models differentiate with one model trained with only TD data and the second model with both TD and CP data. Additionally addressing the lack of childlike free play activities in existing models and if deviating movement patterns in children is of significance for model performance. Furthermore, this study investigated the impact of window size and overlapping versus non-overlapping windows on prediction accuracy, and consider practical considerations related to time and storage use.

## 2. Materials and methods

### 2.1 Participants

Data from 63 TD children and 16 children with CP are included in this validation study (Table 1). The TD children were recruited through a local primary and junior high school, and through colleagues and friends. The children with CP were recruited during a habilitation stay or through outpatient clinic at the local hospital. Only children classified with Gross Motor Function Classification System (GMFCS) I and II were included [40]. All participants were given written age-appropriate information about the study. This was distributed at school and taken home together with written information and consent forms to their parents/guardians. If the parents gave their written consent, verbal information was given to the children before the study started. The data collection started 19.10.2016 and ended 16.08.2019. The study was approved by the Norwegian Centre for Research Data (NSD-nr:50683). In addition, the study was reported to the Regional Ethical Committee for Medical and Health science (REK-nr:2016/707/REK nord) but was not classified under the act on medical and health research. The medical background information that was collected was age, gender, and presence of CP and GMFCS level.

### 2.2 Validation protocols and test groups

The children conducted standardised semi-structured activities including different modes of running, walking, standing, sitting, and lying down, with varying durations. In addition, they performed a free-living protocol including ball games and free play, conducted both indoors and outdoors. For data-synchronisation purposes, the children also performed heel-drops, or the researcher flicked the accelerometer three times. When testing the HAR models, we

**Table 1. Participant characteristics.**

| Subjects | TD | CP |
|---|---|---|
| N (B/G) | 63 (35/28) | 16 (8/8) |
| Age (years) | 10.5 (+/-2.6) [6–15] | 11.4 (+/- 2.2) [8–17] |
| Height (cm) | 149.1 (+/-15.7) [117–170] | 146.7 (+/-10.8) [129–173] |
| Weight (kg) | 42.7 (+/13.0) [21–76] | 43.2 (+/-11.0) [25–62] |
| GMFCS | | |
| I | | 10 |
| II | | 6 |

[1] Mean with (+/- SD) and [range]. TD = Typically developing, CP = Cerebral Palsy, N = Number of subjects, B = Boys, G = Girls, cm = Centimetre, kg = Kilogram, GMFCS = Gross motor function classification scale.

divided the participants into three groups, based on activity protocol and if they were diagnosed with CP. The TD group included the typically developing children who completed both standardised activities and five minutes of free play. One CP group, here after called CP Stan, are children with CP who completed the same standardised activities as the TD group, except for the five minutes of free play. The second CP group, here after called CP Free, included children with CP who only engaged in group free play activities. For both CP and TD, all activities were performed in a single session, and all TD participants conducted the whole protocol. See Supporting information (S1 Table) for full list of activities and number of children conducting the different activities.

## 2.3 Instrumentation

**2.3.1 Activity monitors.** All participants wore two Axivity AX3 accelerometers (Axivity Ltd, Newcastle, UK), one on the thigh placed along the anterior midline, in the middle between anterior superior iliac spine and proximal patella, and one at the approximate placement of L3. For the CP group the thigh accelerometers were placed on the least affected side. Acceleration was sampled at 100 Hz and 200 Hz for TD and 100 Hz for CP (range ± 8g).

**2.3.2 Video recordings.** Video recording using GoPro Hero 3+ cameras were used to identify the performed activities. The cameras were mounted in corners of the room during inside protocols, play and group activities. For activities with longer duration and/outside activities the GoPro camera was attached with a chest harness, pointing downwards to detect leg movement, or handheld by researcher. The recordings were sampled at 60 frames per second, resolution of 1080x720 pixels.

## 2.4 Data processing

**2.4.1 Video annotation.** The video recordings were used as the ground truth for activity types. The activities in the videos were manually labelled (annotated) frame by frame for each participant using Anvil video annotation tool (version 6) [41]. Thirteen activities were labelled using activity definitions used in previous validation studies with the NTNU-HAR models [7, 13, 14]. After clarifying the activity definitions and discussions based on video examples four raters annotated each video independently. Inter-rater reliability of > 0.95 on this methodology has been reported in earlier studies using the same methods, activity definitions and overlapping raters with the present study [7, 14]. Definitions of activities are listed in S2 Table.

**2.4.2 Data pre-processing and feature extraction.** Before training our machine learning model, we performed three pre-processing steps (Fig 1). Initially we down sampled and synchronized thigh and back accelerometer signals with activity annotations ensuring data alignment at the recommended 50 Hz [12]. Subsequently, we segmented the signals into signal frames of our selected window sizes (1 sec, 3 sec, and 5 sec), with and without 50% overlap. The overlap was included to investigate potential loss of activities at the endpoint of the window. Majority voting was applied to the annotations, such that each 1, 3, or 5 second signal frames corresponded to exactly one activity, based on the most frequently occurring in the set window. Lastly, we computed 161 time- and frequency-domain features for each signal frame, using the movement and gravitational components of all six sensor axes, and each sensor's vector magnitude, as described in Logacjov et al., [13]. These resulting features and annotations were used to train the machine learning models.

**2.4.3 Machine learning approach.** We used the Extreme gradient boost (XGBoost) classifier [42] as our machine learning method.

The XGBoost is an ensemble learning approach based on the gradient boosting algorithm [43], where multiple weak classifiers (e.g., decision trees) are trained in a sequential manner.

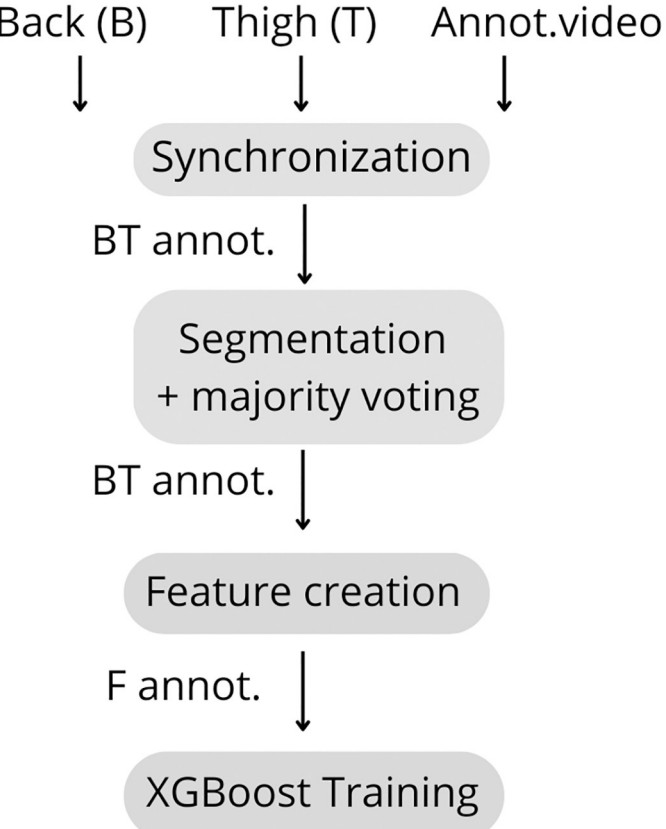

**Fig 1. The pre-processing and feature extraction.**

Each weak classifier was trained to minimize the errors made by the previous weak classifier. Our final model prediction is the weighted sum of all weak classifiers' predictions.

**2.4.4 Model training and cross validation (evaluation).** The XGBoost methodology resulted in two HAR models, NTNU-HAR-Children (HAR-Children) and NTNU-HAR-ChildrenCP (HAR-ChildrenCP). The HAR-Children were trained with data from the TD group and the HAR-ChildrenCP was trained with all TD training data and additional data from children with CP. The two main models comprise a total of 12 sub-models, 6 for each main model, covering 1-, 3-, and 5-seconds windows with and without overlapping windows, see Supporting information (S1 Fig) for overview of the 12 sub-model creation. For each model we initially performed a leave-six-subjects-out-cross-validation with hyperparameter optimization in the form of a grid search. This allowed us to find optimal hyperparameters for each of the 12 sub-models, leading to a fairer comparison. After finding optimal hyperparameters for each model, we performed 12 leave-one-subject-out cross-validations (LOSOCV), one for each model. In the LOSOCV, the model was trained on all participants, except for one, and this participant became the test-data. The overall performance of the model was estimated by repeating this process for each individual in the dataset and then averaging the performance across each individual [13, 16]. This gave us less subject-dependent estimates and thereby less subject-based bias [13, 16, 44]. Note that the six HAR-Children models were trained without the CP data. Hence, the performed LOSOCVs only provided test results for the TD group in this model. To get the results for the CP groups from the HAR-Children models we trained

the model on the whole TD dataset, and then used the CP data as test data. Additionally, before we compared the results of the different window sizes (1 sec, 3 sec, 5 sec), we unfolded the model predictions to the original 50 samples per second, to make all our models comparable. The complete dataset and model file is available at Dataverse (https://doi.org/10.18710/EPCXCC) and GitHub (https://github.com/ntnu-ai-lab/harth-ml-experiments), and is named NTNU-Children.

## 2.5 Data post processing

Initially we annotated with all activity definitions provided in supporting information (S2 Table), we choose to do this to provide precision and to avoid confusing the model when training it on with similar movements. For the further processing, we collapsed some activity labels to make it applicable for practical use. We have defined shuffling as standing with small foot movements (S2 Table), and for our current focus the differentiation between standing still and standing with some foot movement is of limited significance. Therefore, shuffling was imbedded into standing. Similarly, bending is an activity that typically occur when standing and is therefore collapsed with standing. To avoid confusing our model's ability to recognize level walking, walking up and down stairs was collapsed with walking. The original two categories of cycling, sit cycling and stand cycling were collapsed into cycling, as our primary interest was in recognize the cyclic leg movements. These collapsed activity classes are the same as used in previous NTNU-HAR models [7, 13, 14]. The overall preliminary results are provided in S3 Table., in their originally annotated form.

## 2.6 Statistical analysis

We assessed the HAR models by calculating the overall accuracy for each test group and determining precision, sensitivity, specificity, and F1 Score for each activity type. Sensitivity, also called recall, measured our model's ability to correctly classify activities when they occurred, while specificity evaluated the ability to avoid false recognition when activities were absent. Precision indicated the ratio of correctly classified activities to the sum of correctly and falsely classified activities. The F1 Score, a harmonic mean between precision and sensitivity, provided a class-wise precision and sensitivity measure. Accuracy was calculated as the ratio of correctly recognized activity samples to the total number of activity samples. These metrics range from 0–1, with higher values indicating superior performance. The confusion matrixes include the same collapsed activity classes. If the subject did not conduct the activity, they were taken out of the average calculations. We performed all these calculations for each subject before calculating group mean and confidence intervals. All calculations were executed in MATLAB.

## 3. Results

Fig 2 shows the distribution of the performed activities in the three test groups. There was more data from TD children (total 2997.9 minutes) than CP, and more with CP standardised (total 647.4 minutes) than CP Free (total 180 minutes). In all three test groups there were more time spent with walking, standing, sitting, and running than the other activity labels (Fig 2).

## 3.1 Overall performance of the twelve HAR-models

For all models the overall accuracy was high, with the same median value (0.93) (Table 2), and the accuracy slightly favoured (range: 0.12–0.18) the TD and CP Stan groups compared to CP Free. The 1-second model in both HAR-Children and HAR-ChildrenCP performed exemplary

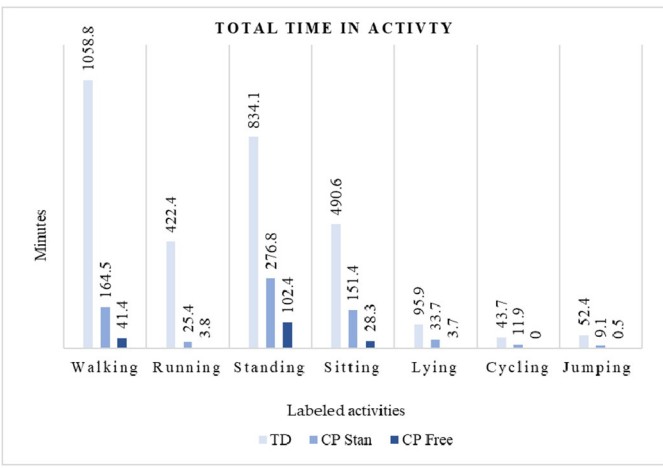

**Fig 2. Bar plot of total amount of activity in the three test groups.** TD = Typically developing children, CP Stan = Children with CP, standardised activities, CP Free = Children with CP, free play.

for the respective test groups (Table 2). The largest difference in accuracy was between the corresponding HAR-Children and HAR-ChildrenCP models for the test group CP Free (range: 0.2–0.4, see Table 2). As illustrated in Table 2 the difference between overlap and non-overlap is only present in HAR-Children 1- and 3-seconds, and from this point on we present the non-overlapping models. Non-overlapping models are preferred for their efficiency in processing time and storage, making them more practical for later use. Accuracy for the original activity classes is presented in S3 Table. All experiment results can be found at GITHUB (https://github.com/ntnu-ai-lab/harth-ml-experiments).

## 3.2 Specific activity performance of the six selected HAR-models

In both 1-second models (Table 3a and 3b) all activities were predicted with high accuracy for the TD group, with F1 Score over 0.85. In CP Stan only running (range: 0.79–0.80) and cycling (range: 0.75–0.78) had a F1 Score under 0.90. The CP Free group had slightly lower accuracy and a wider range in both 1-second models (range: 0.29–0.94). This group had superior values in favour of the HAR-Children CP model, with an average difference of 0.09 (Table 3b), where walking, running, and jumping had the largest difference (0.09, 0.07, 0.27).

**Table 2. Overall accuracy for each model with and without overlaps.**

| NTNU-HAR-Children | 1 sec | 1 sec overlap | 3 sec | 3 sec overlap | 5 sec | 5 sec Overlap |
|---|---|---|---|---|---|---|
| Test group | | | | | | |
| TD | 0.94 | 0.95 | 0.93 | 0.93 | 0.92 | 0.92 |
| CP Stan | 0.94 | 0.94 | 0.94 | 0.94 | 0.93 | 0.93 |
| CP Free | 0.78 | 0.79 | 0.77 | 0.78 | 0.75 | 0.75 |
| NTNU-HAR ChildrenCP | 1 sec | 1 sec overlap | 3 sec | 3 sec overlap | 5 sec | 5 sec overlap |
| TD | 0.94 | 0.94 | 0.94 | 0.94 | 0.92 | 0.92 |
| CP Stan | 0.95 | 0.95 | 0.94 | 0.94 | 0.93 | 0.93 |
| CP Free | 0.82 | 0.82 | 0.81 | 0.80 | 0.78 | 0.77 |

Test groups: TD = Typically developing children, CP Stan = Cerebral palsy with standardised activities, CP Free = Cerebral palsy with Free play. The range of accuracy is 0–1, where higher scores are better.

**Table 3. Sensitivity, specificity, precision, and F1 Score for each activity.**

| a) | **NTNU HAR-Children- One second** | | | | | |
|---|---|---|---|---|---|---|
| Test group | Activity type | | | | | |
| TD | | Sub | Sensitivity | Specificity | Precision | F1 Score |
| | Walking | 62 | 0.94 [0.92, 0.95] | 0.98 [0.97, 0.98] | 0.95 [0.94, 0.96] | 0.94 [0.93, 0.96] |
| | Running | 62 | 0.96 [0.94, 0.97] | 0.99 [0.99, 0.99] | 0.91 [0.87, 0.94] | 0.92 [0.90, 0.95] |
| | Standing | 62 | 0.93 [0.89, 0.96] | 0.97 [0.97, 0.98] | 0.93 [0.91, 0.95] | 0.92 [0.89, 0.95] |
| | Sitting | 62 | 0.98 [0.97, 0.99] | 0.99 [0.97, 1.00] | 0.97 [0.94, 0.99] | 0.97 [0.95, 0.99] |
| | Lying | 29 | 0.99 [0.98, 1.00] | 1.00 [0.99, 1.00] | 0.98 [0.94, 1.01] | 0.98 [0.95, 1.00] |
| | Cycling | 21 | 0.87 [0.78, 0.97] | 1.00 [1.00, 1.00] | 0.93 [0.87, 0.98] | 0.89 [0.81, 0.97] |
| | Jumping | 17 | 0.87 [0.77, 0.96] | 1,00 [1.00, 1.00] | 0.92 [0.86, 0.98] | 0.87 [0.79, 0.95] |
| CP Stan | | Sub | Sensitivity | Specificity | Precision | F1 Score |
| | Walking | 10 | 0.89 [0.86, 0.93] | 0.97 [0.96, 0.98] | 0.92 [0.90, 0.94] | 0.90 [0.88, 0.93] |
| | Running | 10 | 0.77 [0.60, 0.94] | 1.00 [0.99, 1.00] | 0.88 [0.83, 0.93] | 0.80 [0.66, 0.94] |
| | Standing | 10 | 0.95 [0.93, 0.97] | 0.95 [0.93, 0.96] | 0.92 [0.89, 0.94] | 0.93 [0.91, 0.95] |
| | Sitting | 10 | 0.99 [0.99, 1.00] | 1.00 [0.99, 1.00] | 0.99 [0.98, 1.00] | 0.99 [0.99, 0.99] |
| | Lying | 10 | 0.99 [0.98, 1.00] | 1.00 [1.00, 1.00] | 0.99 [0.98, 1.00] | 0.99 [0.98, 1.00] |
| | Cycling | 2 | 0.62 [-1.37, 2.61] | 1.00 [1.00, 1.00] | 0.99 [0.92, 1.07] | 0.75 [-0.80, 2.30] |
| | Jumping | 7 | 0.91 [0.85, 0.98] | 1.00 [1.00, 1.00] | 0.95 [0.91, 0.99] | 0.93 [0.88, 0.98] |
| CP Free | | Sub | Sensitivity | Specificity | Precision | F1 Score |
| | Walking | 6 | 0.49 [0.34, 0.64] | 0.92 [0.90, 0.95] | 0.65 [0.59, 0.70] | 0.55 [0.45, 0.65] |
| | Running | 5 | 0.60 [0.46, 0.73] | 0.99 [0.98, 1.00] | 0.52 [0.20, 0.84] | 0.53 [0.29, 0.77] |
| | Standing | 6 | 0.88 [0.84, 0.92] | 0.70 [0.59, 0.82] | 0.80 [0.75, 0.84] | 0.84 [0.81, 0.86] |
| | Sitting | 6 | 0.87 [0.75, 0.99] | 0.99 [0.98, 1.00] | 0.93 [0.86, 1.00] | 0.90 [0.82, 0.97] |
| | Lying | 6 | 0.89 [0.77, 1.01] | 0.99 [0.95, 1.02] | 0.71 [0.29, 1.14] | 0.74 [0.38, 1.11] |
| | Jumping | 2 | 0.19 [-0.03, 0.40] | 1.00 [1.00, 1.00] | 0.72 [-2.91, 4.34] | 0.29 [-0.31, 0.90] |
| b) | **NTNU HAR-ChildrenCP- One second** | | | | | |
| Test group | Activity type | | | | | |
| TD | | Sub | Sensitivity | Specificity | Precision | F1 Score |
| | Walking | 62 | 0.94 [0.92, 0.96] | 0.97 [0.97, 0.98] | 0.95 [0.94, 0.96] | 0.94 [0.93, 0.96] |
| | Running | 62 | 0.96 [0.95, 0.97] | 0.99 [0.99, 0.99] | 0.90 [0.87, 0.94] 0.94 [0.92, 0.95] | 0.92 [0.90, 0.95] |
| | Standing | 61 | 0.94 [0.93, 0.95] | 0.98 [0.97, 0.98] | 0.94 [0.92, 0.95] 0.97 [0.94, 0.99] | 0.94 [0.93, 0.95] |
| | Sitting | 62 | 0.98 [0.97, 0.99] 0.99 [0.99, 1.00] | 0.99 [0.96, 1.01] | 0.97 [0.94, 0.99] | 0.97 [0.95, 0.99] |
| | Lying | 29 | 0.99 [0.99, 1.00] | 1.00 [1.00, 1.00] | 0.98 [0.95, 1.01] | 0.98 [0.97, 1.00] |
| | Cycling | 21 | 0.90 [0.82, 0.97] | 1.00 [1.00, 1.00] | 0.95 [0.92, 0.98] | 0.92 [0.86, 0.98] |
| | Jumping | 17 | 0.83 [0.72, 0.93] | 1.00 [1.00, 1.00] | 0.91 [0.84, 0.98] | 0.86 [0.77, 0.94] |
| CP Stan | | Sub | Sensitivity | Specificity | Precision | F1 Score |
| | Walking | 10 | 0.91 [0.88, 0.94] | 0.97 [0.96, 0.98] | 0.91 [0.89, 0.94] 0.87 [0.80, 0.94] | 0.91 [0.89, 0.93] |
| | Running | 10 | 0.76 [0.57, 0.95] | 1.00 [0.99, 1.00] | 0.87 [0.80, 0.94] | 0.79 [0.63, 0.95] |
| | Standing | 10 | 0.95 [0.93, 0.97] | 0.96 [0.94, 0.97] 1.00 [1.00, 1.00] | 0.93 [0.91, 0.96] | 0.94 [0.92, 0.96] |
| | Sitting | 10 | 0.99 [0.99, 1.00] | 1.00 [1.00, 1.00] | 0.99 [0.98, 1.00] | 0.99 [0.99, 1.00] |
| | Lying | 10 | 0.99 [0.98, 1.00] | 1.00 [1.00, 1.00] | 0.99 [0.98, 1.00] | 0.99 [0.98, 1.00] 0.78 [-0.88, 2.44] |
| | Cycling | 2 | 0.66 [-1.59, 2.91] 0.93 [0.86, 1.00] | 1.00 [1.00, 1.00] | 1.00 [1.00, 1.00] | 0.78 [-0.88, 2.44] |
| | Jumping | 7 | 0.93 [0.86, 1.00] | 1.00 [1.00, 1.00] | 0.95 [0.92, 0.98] | 0.94 [0.89, 0.99] |

*(Continued)*

**Table 3.** (*Continued*)

| CP Free | | Sub | Sensitivity | Specificity | Precision | F1 Score |
|---|---|---|---|---|---|---|
| | Walking | 6 | 0.61 [0.47, 0.75] | 0.92 [0.89, 0.95] 0.99 [0.98, 1.00] | 0.69 [0.62, 0.76] | 0.64 [0.57, 0.71] |
| | Running | 6 | 0.56 [0.44, 0.67] | 0.99 [0.98, 1.00] | 0.45 [0.14, 0.76] | 0.46 [0.22, 0.71] |
| | Standing | 6 | 0.89 [0.85, 0.93] | 0.79 [0.68, 0.90] | 0.85 [0.81, 0.90] | 0.87 [0.86, 0.88] |
| | Sitting | 6 | 0.92 [0.80, 1.04] | 1.00 [0.99, 1.00] | 0.96 [0.93, 1.00] | 0.94 [0.87, 1.00] |
| | Lying | 6 | 0.88 [0.78, 0.98] | 0.99 [0.95, 1.02] | 0.76 [0.38, 1.14] | 0.76 [0.45, 1.07] |
| | Jumping | 3 | 0.43 [-0.14, 1.01] | 1.00 [1.00, 1.00] | 0.84 [0.15, 1.53] | 0.56 [-0.04, 1.16] |
| **c)** | **NTNU HAR-Children- Three seconds** | | | | | |
| Test group | Activity type | | | | | |
| TD | | Sub | Sensitivity | Specificity | Precision | F1 Score |
| | Walking | 62 | 0.93 [0.92, 0.95] 0.94 [0.93, 0.96] | 0.96 [0.96, 0.97] | 0.94 [0.92, 0.95] | 0.93 [0.92, 0.95] |
| | Running | 62 | 0.94 [0.93, 0.96] | 0.99 [0.98, 0.99] | 0.89 [0.85, 0.92] | 0.91 [0.88, 0.93] |
| | Standing | 62 | 0.90 [0.87, 0.93] | 0.97 [0.97, 0.98] | 0.93 [0.92, 0.95] | 0.91 [0.87, 0.94] |
| | Sitting | 62 | 0.97 [0.96, 0.98] | 0.98 [0.97, 1.00] | 0.96 [0.94, 0.98] | 0.96 [0.94, 0.98] |
| | Lying | 29 | 0.98 [0.97, 0.99] | 0.99 [0.98, 1.00] | 0.96 [0.90, 1.01] 0.93 [0.87, 0.99] | 0.96 [0.91, 1.00] |
| | Cycling | 21 | 0.88 [0.79, 0.97] | 1.00 [1.00, 1.00] | 0.93 [0.87, 0.99] | 0.90 [0.82, 0.98] |
| | Jumping | 15 | 0.92 [0.88, 0.96] | 1.00 [1.00, 1.00] | 0.91 [0.83, 1.00] | 0.90 [0.85, 0.96] |
| CP Stan | | Sub | Sensitivity | Specificity | Precision | F1 Score |
| | Walking | 10 | 0.90 [0.87, 0.93] | 0.97 [0.95, 0.98] | 0.90 [0.87, 0.93] | 0.90 [0.87, 0.93] |
| | Running | 9 | 0.78 [0.69, 0.88] | 1.00 [0.99, 1.00] | 0.90 [0.85, 0.96] | 0.83 [0.77, 0.89] |
| | Standing | 10 | 0.94 [0.91, 0.97] | 0.95 [0.94, 0.96] | 0.92 [0.89, 0.95] | 0.93 [0.90, 0.96] |
| | Sitting | 10 | 0.99 [0.99, 1.00] | 0.99 [0.99, 1.00] | 0.98 [0.96, 0.99] | 0.98 [0.98, 0.99] |
| | Lying | 10 | 0.99 [0.98, 1.00] | 1.00 [1.00, 1.00] | 0.99 [0.98, 1.00] | 0.99 [0.98, 1.00] |
| | Cycling | 2 | 0.64 [0.11, 1.17] | 1.00 [1.00, 1.00] | 0.99 [0.92, 1.07] | 0.78 [0.41, 1.14] |
| | Jumping | 7 | 0.84 [0.71, 0.97] | 1.00 [1.00, 1.00] | 0.98 [0.95, 1.00] | 0.90 [0.81, 0.98] |
| CP Free | | Sub | Sensitivity | Specificity | Precision | F1 Score |
| | Walking | 6 | 0.58 [0.40, 0.75] | 0.88 [0.84, 0.92] | 0.59 [0.54, 0.63] | 0.57 [0.48, 0.66] |
| | Running | 5 | 0.49 [0.19, 0.79] | 0.98 [0.96, 1.00] | 0.43 [0.11, 0.75] | 0.41 [0.15, 0.67] |
| | Standing | 6 | 0.83 [0.78, 0.89] | 0.75 [0.62, 0.88] | 0.82 [0.77, 0.87] | 0.82 [0.79, 0.86] |
| | Sitting | 6 | 0.86 [0.75, 0.97] | 0.99 [0.98, 1.00] | 0.91 [0.82, 1.01] | 0.88 [0.81, 0.95] |
| | Lying | 6 | 0.87 [0.75, 1.00] | 0.99 [0.95, 1.02] | 0.71 [0.31, 1.11] | 0.74 [0.39, 1.09] |
| | Jumping | 0 | 0.00 [0.00, 0.00] | 1.00 [1.00, 1.00] | 0.00 [0.00, 0.00] | 0.00 [0.00, 0.00] |
| **d)** | **NTNU HAR-ChildrenCP- Three seconds** | | | | | |
| Test group | Activity type | | | | | |
| TD | | Sub | Sensitivity | Specificity | Precision | F1 Score |
| | Walking | 62 | 0.94 [0.92, 0.95] 0.95 [0.94, 0.96] | 0.97 [0.96, 0.97] | 0.94 [0.93, 0.95] | 0.94 [0.93, 0.95] |
| | Running | 62 | 0.95 [0.94, 0.96] | 0.99 [0.98, 0.99] | 0.89 [0.86, 0.93] | 0.91 [0.89, 0.94] |
| | Standing | 62 | 0.90 [0.87, 0.93] | 0.97 [0.97, 0.98] | 0.93 [0.92, 0.95] | 0.91 [0.87, 0.94] |
| | Sitting | 62 | 0.97 [0.96, 0.98] | 0.98 [0.97, 1.00] | 0.96 [0.93, 0.98] | 0.96 [0.94, 0.98] |
| | Lying | 29 | 0.98 [0.97, 0.99] | 1.00 [1.00, 1.00] | 0.97 [0.95, 1.00] | 0.97 [0.96, 0.99] |
| | Cycling | 20 | 0.93 [0.88, 0.97] | 1.00 [1.00, 1.00] | 0.94 [0.88, 1.00] | 0.93 [0.88, 0.98] |
| | Jumping | 14 | 0.92 [0.88, 0.95] | 1.00 [1.00, 1.00] | 0.95 [0.93, 0.97] | 0.93 [0.91, 0.95] |
| CP Stan | | Sub | Sensitivity | Specificity | Precision | F1 Score |
| | Walking | 10 | 0.91 [0.89, 0.94] | 0.96 [0.95, 0.98] | 0.90 [0.86, 0.93] | 0.90 [0.88, 0.93] |
| | Running | 9 | 0.79 [0.67, 0.90] | 1.00 [0.99, 1.00] | 0.89 [0.82, 0.95] | 0.82 [0.76, 0.89] |

(*Continued*)

**Table 3.** (Continued)

| | | Sub | Sensitivity | Specificity | Precision | F1 Score |
|---|---|---|---|---|---|---|
| | Standing | 10 | 0.93 [0.90, 0.96] | 0.95 [0.94, 0.97] | 0.93 [0.90, 0.96] | 0.93 [0.90, 0.96] |
| | Sitting | 10 | 0.99 [0.99, 0.99] | 1.00 [0.99, 1.00] | 0.98 [0.97, 0.99] | 0.99 [0.98, 0.99] |
| | Lying | 10 | 0.99 [0.98, 1.00] | 1.00 [1.00, 1.00] | 0.99 [0.98, 1.00] 1.00 [1.00, 1.00] | 0.99 [0.98, 1.00] |
| | Cycling | 2 | 0.66 [0.56, 0.75] | 1.00 [1.00, 1.00] | 1.00 [1.00, 1.00] | 0.79 [0.72, 0.86] |
| | Jumping | 7 | 0.86 [0.76, 0.96] | 1.00 [1.00, 1.00] | 0.96 [0.92, 1.01] | 0.91 [0.83, 0.98] |
| CP Free | | Sub | Sensitivity | Specificity | Precision | F1 Score |
| | Walking | 6 | 0.64 [0.49, 0.79] | 0.90 [0.87, 0.93] | 0.65 [0.58, 0.71] | 0.63 [0.57, 0.69] |
| | Running | 5 | 0.54 [0.33, 0.74] | 0.99 [0.97, 1.00] | 0.48 [0.29, 0.68] | 0.49 [0.31, 0.68] |
| | Standing | 6 | 0.86 [0.80, 0.91] | 0.79 [0.68, 0.91] | 0.85 [0.80, 0.91] | 0.85 [0.84, 0.87] |
| | Sitting | 6 | 0.91 [0.79, 1.03] | 1.00 [0.99, 1.00] | 0.96 [0.93, 1.00] | 0.93 [0.86, 1.00] |
| | Lying | 6 | 0.84 [0.68, 0.99] | 0.99 [0.95, 1.02] | 0.72 [0.32, 1.12] | 0.72 [0.38, 1.05] |
| | Jumping | 0 | 0.00 [0.00, 0.00] | 1.00 [1.00, 1.00] | 0.00 [0.00, 0.00] | 0.00 [0.00, 0.00] |
| **e)** | **NTNU HAR-Children- Five seconds** | | | | | |
| Test group | Activity type | | | | | |
| TD | | Sub | Sensitivity | Specificity | Precision | F1 Score |
| | Walking | 62 | 0.93 [0.91, 0.94] | 0.96 [0.95, 0.97] | 0.92 [0.91, 0.94] | 0.92 [0.91, 0.94] |
| | Running | 62 | 0.93 [0.91, 0.95] | 0.99 [0.98, 0.99] | 0.87 [0.83, 0.91] | 0.89 [0.86, 0.92] |
| | Standing | 62 | 0.87 [0.84, 0.91] | 0.97 [0.96, 0.98] | 0.92 [0.90, 0.93] | 0.89 [0.85, 0.92] |
| | Sitting | 62 | 0.96 [0.95, 0.98] | 0.98 [0.97, 1.00] | 0.95 [0.92, 0.97] | 0.95 [0.93, 0.97] |
| | Lying | 29 | 0.97 [0.95, 0.98] | 0.99 [0.99, 1.00] | 0.94 [0.89, 0.99] | 0.95 [0.91, 0.99] |
| | Cycling | 21 | 0.88 [0.80, 0.96] | 1.00 [1.00, 1.00] | 0.93 [0.88, 0.97] | 0.89 [0.82, 0.96] |
| | Jumping | 14 | 0.92 [0.89, 0.96] | 1.00 [1.00, 1.00] | 0.94 [0.92, 0.97] | 0.93 [0.91, 0.95] |
| CP Stan | | Sub | Sensitivity | Specificity | Precision | F1 Score |
| | Walking | 10 | 0.89 [0.86, 0.92] | 0.96 [0.95, 0.98] | 0.89 [0.85, 0.93] | 0.89 [0.85, 0.92] |
| | Running | 9 | 0.78 [0.65, 0.92] | 0.99 [0.99, 1.00] | 0.84 [0.79, 0.90] | 0.80 [0.71, 0.88] |
| | Standing | 10 | 0.92 [0.88, 0.96] 0.99 [0.98, 0.99] | 0.95 [0.93, 0.96] | 0.91 [0.87, 0.95] | 0.92 [0.88, 0.95] |
| | Sitting | 10 | 0.99 [0.98, 0.99] | 0.99 [0.99, 1.00] | 0.97 [0.96, 0.99] | 0.98 [0.97, 0.99] |
| | Lying | 10 | 0.99 [0.98, 1.00] | 1.00 [1.00, 1.00] | 0.98 [0.96, 0.99] | 0.98 [0.97, 0.99] |
| | Cycling | 2 | 0.59 [-0.74, 1.92] | 1.00 [1.00, 1.00] | 0.99 [0.80, 1.17] | 0.73 [-0.37, 1.83] 0.87 [0.77, 0.97] |
| | Jumping | 7 | 0.83 [0.67, 0.98] | 1.00 [1.00, 1.00] | 0.95 [0.89, 1.00] | 0.87 [0.77, 0.97] |
| CP Free | | Sub | Sensitivity | Specificity | Precision | F1 Score |
| | Walking | 6 | 0.59 [0.42, 0.77] | 0.87 [0.82, 0.93] | 0.58 [0.52, 0.65] | 0.57 [0.48, 0.66] |
| | Running | 5 | 0.52 [0.24, 0.80] | 0.98 [0.96, 0.99] | 0.33 [0.12, 0.55] | 0.37 [0.19, 0.54] |
| | Standing | 6 | 0.81 [0.73, 0.89] | 0.75 [0.59, 0.90] | 0.82 [0.76, 0.88] | 0.81 [0.76, 0.85] |
| | Sitting | 6 | 0.82 [0.69, 0.96] | 0.99 [0.98, 1.00] | 0.89 [0.75, 1.03] | 0.85 [0.74, 0.95] |
| | Lying | 6 | 0.91 [0.83, 0.99] | 0.98 [0.95, 1.02] | 0.65 [0.26, 1.05] | 0.70 [0.36, 1.04] |
| | Jumping | 0 | 0.00 [0.00, 0.00] | 1.00 [1.00, 1.00] | 0.00 [0.00, 0.00] | 0.00 [0.00, 0.00] |
| **f)** | **NTNU HAR-ChildrenCP- Five seconds** | | | | | |
| Test group | Activity type | | | | | |
| TD | | Sub | Sensitivity | Specificity | Precision | F1 Score |
| | Walking | 62 | 0.93 [0.92, 0.95] | 0.96 [0.95, 0.97] | 0.92 [0.91, 0.94] | 0.93 [0.91, 0.94] |
| | Running | 62 | 0.94 [0.92, 0.96] | 0.99 [0.98, 0.99] | 0.87 [0.83, 0.90] | 0.90 [0.87, 0.92] |
| | Standing | 61 | 0.89 [0.86, 0.91] | 0.97 [0.97, 0.98] | 0.92 [0.91, 0.94] | 0.90 [0.88, 0.92] |
| | Sitting | 62 | 0.97 [0.96, 0.98] | 0.98 [0.96, 1.00] | 0.95 [0.92, 0.97] | 0.95 [0.94, 0.97] |
| | Lying | 29 | 0.98 [0.97, 0.99] | 1.00 [0.99, 1.00] | 0.95 [0.92, 0.98] | 0.96 [0.94, 0.98] |
| | Cycling | 21 | 0.90 [0.84, 0.97] | 1.00 [1.00, 1.00] | 0.91 [0.86, 0.96] | 0.90 [0.84, 0.96] |

(*Continued*)

**Table 3.** (Continued)

| | | Sub | Sensitivity | Specificity | Precision | F1 Score |
|---|---|---|---|---|---|---|
| | Jumping | 14 | 0.92 [0.88, 0.96] | 1.00 [1.00, 1.00] | 0.93 [0.90, 0.96] | 0.92 [0.90, 0.94] |
| CP Stan | | Sub | Sensitivity | Specificity | Precision | F1 Score |
| | Walking | 10 | 0.90 [0.88, 0.93] | 0.96 [0.94, 0.98] 0.99 [0.99, 1.00] | 0.89 [0.84, 0.93] | 0.90 [0.86, 0.93] |
| | Running | 9 | 0.82 [0.70, 0.94] | 0.99 [0.99, 1.00] | 0.85 [0.80, 0.91] | 0.83 [0.75, 0.91] |
| | Standing | 10 | 0.92 [0.87, 0.97] | 0.95 [0.94, 0.96] | 0.92 [0.89, 0.95] | 0.92 [0.88, 0.96] |
| | Sitting | 10 | 0.99 [0.98, 1.00] | 0.99 [0.99, 1.00] | 0.97 [0.96, 0.99] | 0.98 [0.98, 0.99] |
| | Lying | 10 | 0.98 [0.97, 1.00] | 1.00 [1.00, 1.00] | 0.99 [0.98, 1.00] | 0.99 [0.98, 0.99] |
| | Cycling | 2 | 0.70 [0.63, 0.77] | 1.00 [1.00, 1.00] | 1.00 [1.00, 1.00] | 0.82 [0.78, 0.87] |
| | Jumping | 7 | 0.84 [0.70, 0.98] | 1.00 [1.00, 1.00] | 0.95 [0.90, 1.00] | 0.88 [0.79, 0.97] |
| CP Free | | Sub | Sensitivity | Specificity | Precision | F1 Score |
| | Walking | 6 | 0.59 [0.37, 0.80] 0.58 [0.24, 0.92] | 0.89 [0.82, 0.95] | 0.61 [0.52, 0.71] | 0.57 [0.47, 0.67] |
| | Running | 5 | 0.58 [0.24, 0.92] 0.83 [0.74, 0.91] | 0.98 [0.95, 1.00] | 0.38 [0.24, 0.53] | 0.41 [0.28, 0.54] |
| | Standing | 6 | 0.83 [0.74, 0.91] | 0.78 [0.64, 0.91] | 0.84 [0.78, 0.90] | 0.83 [0.81, 0.85] |
| | Sitting | 6 | 0.89 [0.77, 1.01] | 0.99 [0.99, 1.00] | 0.90 [0.76, 1.05] | 0.89 [0.78, 1.00] |
| | Lying | 6 | 0.83 [0.62, 1.05] | 0.99 [0.95, 1.02] | 0.73 [0.36, 1.10] | 0.72 [0.40, 1.05] |
| | Jumping | 0 | 0.00 [0.00, 0.00] | 1.00 [1.00, 1.00] | 0.00 [0.00, 0.00] | 0.00 [0.00, 0.00] |

Sensitivity, specificity, precision, and F1 Score calculated as mean across all participants in the test group and [95% confidence interval]. Sub = Subjects, number of participants detected with the activity. Each sub-table represent one of the HAR-models for the three predefined test groups. TD = Typically developing, CP stan = Cerebral Palsy standardised activities, CP Free = Cerebral Palsy, Free play.

In the two 3-seconds models (Table 3c and 3d) all activities had still high accuracy for TD and CP Stan (range: 0.78–0.99), both the difference from the corresponding 1-second models (average: 0.01–0.02) and between the 3-seconds models (Table 3c and 3d) were small (F1 Score <0.03). For the CP Free group, the HAR-ChildrenCP 3-seconds (Table 3d) had better F1 Scores (diff: 0.02–0.06) than HAR-Children 3-seconds (Table 3c). However, for all activities the HAR-ChildrenCP 1-second model (Table 3b) were slightly better (range: 0.01–0.04) than 3-seconds (Table 3d).

For the TD group the HAR-Children 5-seconds model (Table 3e) had lower F1 Scores in all activities (range of decrease in F1: 0.01–0.09) compared to 1- and 3-seconds models (Table 3a–3d), except for cycling, where it was higher ($<$ 0.06 increase in F1) with the 5-seconds models. In the two CP groups the HAR-ChildrenCP 5-seconds (Table 3f) had higher F1 Scores (range increase in F1: 0.01–0.04) than HAR-Children 5-seconds (Table 3e), except for the activity lying. When comparing the HAR-ChildrenCP 1-second (Table 3b) and the 5-seconds model (Table 3f) for the CP Stan group, the average difference in F1 Score was 0.03. Also, here the cycling score was higher with the 5-seconds model. In CP Free, the average difference in F1 Score was 0.05 in favour of the 1-second model. Mark that this was without jumping, while jumping shows the largest difference (Table 3e and 3f).

## 3.3. Confusion matrixes

In all the confusion matrixes in Figs 3–5 the misclassification was higher in the test group CP Free for all HAR-models, where walking was misclassified as standing on average 37% and standing as walking in 10.8% of the instances. Running was more often misclassified as walking in the two CP groups, with average 34.2% in the CP Free group and 17% in CP Stan.

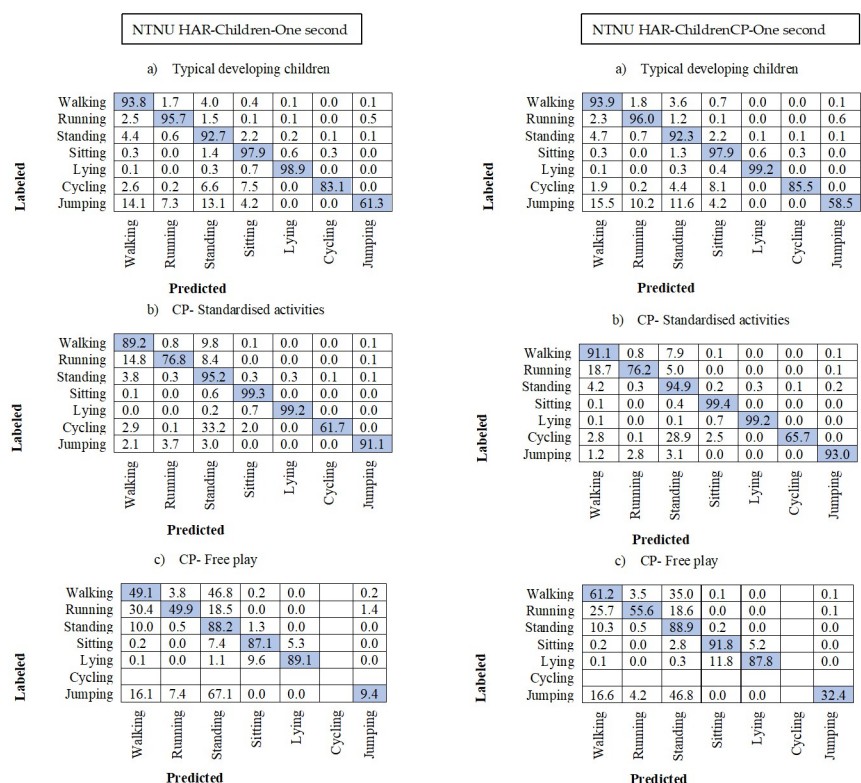

**Fig 3. Confusion matrix for the 1 second models with non-overlapping windows.** One matrix for each of the three test-groups. Number of subjects are the same as presented in Table 3. The rows represent the video annotated or labelled activity types. The columns represent the predicted activity types. All numbers are percentages.

Jumping was often misclassified in CP Free, with 100% misclassification in both 3- and 5-seconds models, however for CP Stan jumping was well predicted for all models and higher than the TD group. The difference between HAR-Children and HAR-ChildrenCP decreased with window size (Figs 3–5). Furthermore, the performance for the HAR-ChildrenCP models decreased with increasing windows while the HAR-Children increase for the CP Free test group, however not for the other test groups (Figs 3–5).

## 4. Discussion

This study validates two HAR models' ability to predict habitual physical activities in TD children and children with CP (GMFCS I & II). Both models accurately predict standardised activities, with the best overall accuracy for the CP Stan group. The two 1-second models outperform the other models for all test groups. The HAR-ChildrenCP 1-second had preferable performance for all test groups in all activities except for cycling, where 3- or 5-seconds perform better. In the CP Free group there was more variability between models and wider confidence intervals within activities. There were also larger differences in F1 Scores between HAR-Children and HAR-ChildrenCP models when tested on the CP Free group. The results of the present study show that window size is of importance, and ideal size depends on the target activity. The most challenging activities for the models to correctly predict are running, walking, and jumping for the CP groups.

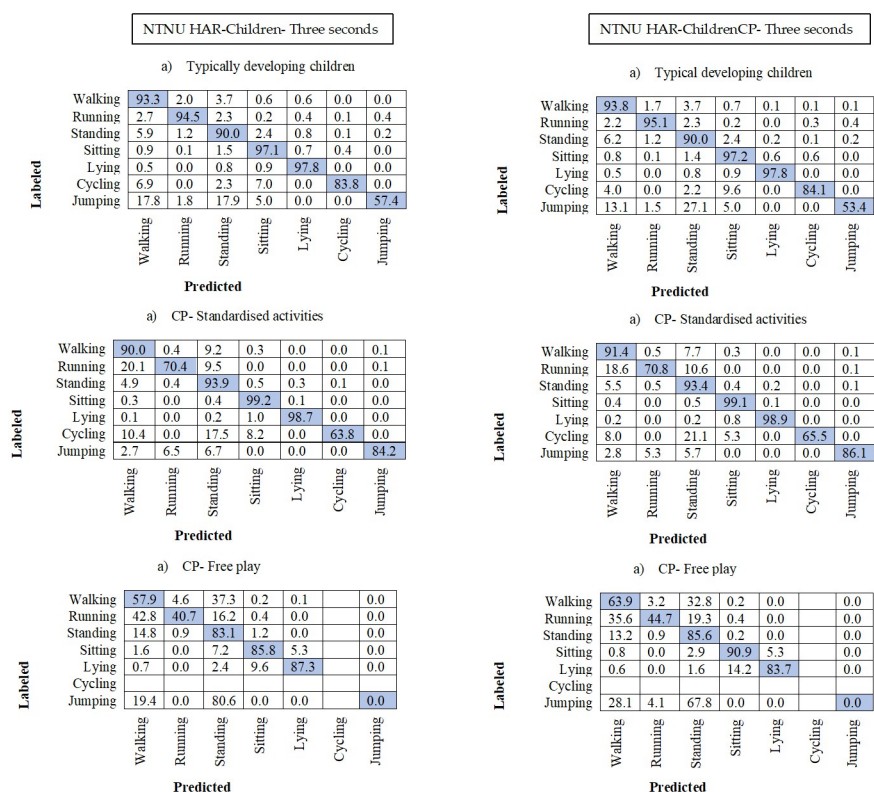

**Fig 4. Confusion matrix for the 3 second models with non-overlapping windows.** One matrix for each of the three test-groups. Number of subjects are the same as presented in Table 3. The rows represent the video annotated or labelled activity types. The columns represent the predicted activity types. All numbers are percentages.

## 4.1 The difference in performance between HAR-Children and HAR-ChildrenCP

The goal of activity recognition is to correctly predict daily life activities, particularly in free-living situations. Our overall accuracy for TD children ranges from 92–95%, representing an improvement over prior models designed for the same age group and simulated free living, which report accuracy between 62–86% [17, 18]. Notably, studies with higher accuracy exists, however tested on a treadmill [16], limiting their applicability to real life situations. The results of the present study align favourably with previous studies including standardised protocols and children with CP, that have achieved accuracy exceeding 90% [24, 33, 34]. Moreover, our study's achievement of accuracies >75% in simulated free-living conditions for children with CP is particularly notable, given the limited existing research with suboptimal accuracy [34]. Our model trained with free play activities also performs best for the CP Free group, highlighting the importance of specific training data [23, 24].

The two models consistently demonstrate superior performance when applied to the TD and CP Stan groups, compared to the CP Free group. Intriguingly, in some models, the CP Stan group achieves higher overall and activity-specific performance than the TD group. This implies that the HAR models may not be primarily challenged by the deviating movement patterns in CP. Instead, the presence of unstandardised activities and sporadic transitions within free play emerges as a potential challenge for HAR models. In our data the inferior performance for the two groups that include free play, TD and CP Free, might also be due to the

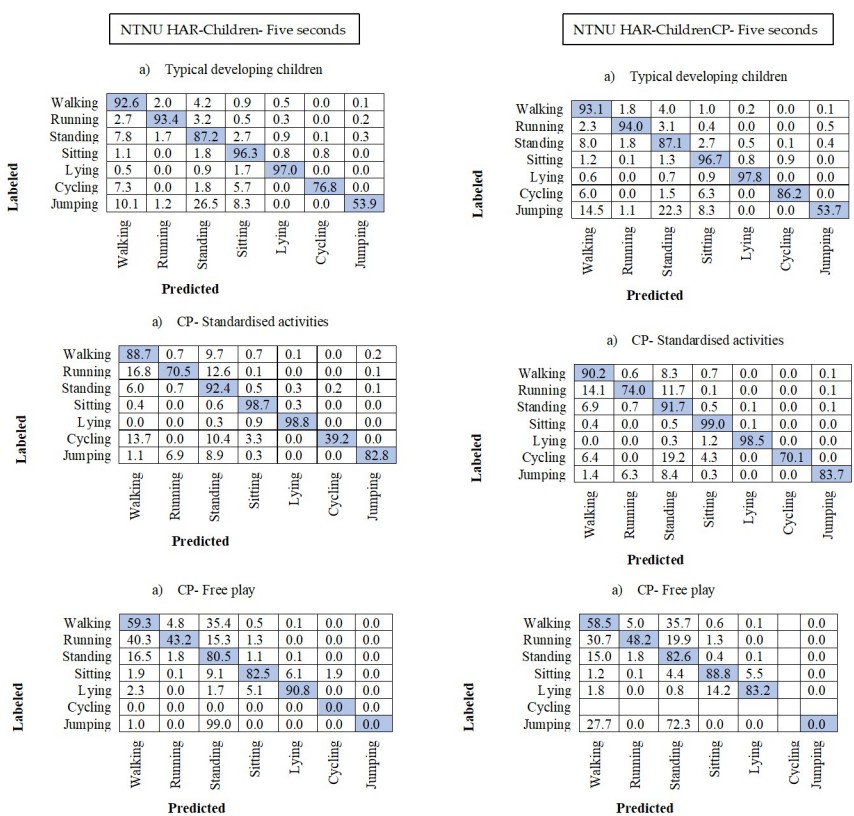

**Fig 5. Confusion matrix for the 5 second models with non-overlapping windows.** One matrix for each of the three test-groups. Number of subjects are the same as presented in Table 3. The rows represent the video annotated or labelled activity types. The columns represent the predicted activity types. All numbers are percentages.

limited training data, with mere five minutes of free play in TD and six participants in CP Free. Nevertheless, free play activities or variability in movement pattens might be beneficial for the model's ability to predict standardised activities, both for TD and children with CP. Which can be exemplified by the superior performance of the HAR-ChildrenCP models for both CP Stan and all activities, except for jumping, in the TD group. This is interesting because one would expect that the heterogeneity within the CP population hinders the model's performance. However, in our experiment, this diversity within the CP population could potentially contribute to improved performance for standardised activities, thereby benefiting both TD and CP Stan. Overall, the HAR-ChildrenCP model includes all 79 children and thereby represent a wide variation of movement patterns, and with the contribution of the CP Free group it offers superior performance in most activities for all test groups.

## 4.2 Accuracy and misclassification of activities

In regards of misclassification of activities, the two groups with standardised activities had a lower misclassification of jumping, compared to free play. We can theorise that structured, sequential jumping, as seen in these protocols, are easier for the models to recognize compared to sporadic jumping or jumping imbedded in other activities in free play. Conversely, even though all jumping from the TD and CP Stan groups were included in the training, it might lack the specificity required for the CP Free group. The misclassification of jumping as

standing and walking was more pronounced in the TD group than the CP Stan group, and thereby supports the notion that the primary challenge is free play activities.

There were better predictions of walking for the TD group compared to the children with CP. This can be attributed to differences in walking patterns between the groups [30, 31]. The asymmetric movement during walking in CP creates a distinct difference in accelerometry from TD children [31]. Furthermore, children with CP require longer time during acceleration, and have a steeper increase in acceleration measures related to increase in speed [31], which might affect the recommended choice of window size. In our results the 5-second HAR-Children model perform best for CP Free, indicating that without CP training data, the models with smaller window sizes have difficulties detecting short bouts of walking and running with different accelerometric features.

The misclassification of running as walking in the two CP groups can also be attributed to asymmetry, which is exacerbated with increasing musculoskeletal demands during running [45]. The misclassification may also be influenced by the representation of different walking and running paces in our training data. In real life scenarios, children tend to transition intermittently between walking and running paces. Therefore, it is crucial for the model to recognize the activities regardless of pace. Notably, this limitation in differentiating walking paces is consistent with other publications on children with CP, none of which include running [24, 33, 34]. Additionally, in existing studies regarding TD children the same misclassifications between walking paces are present, however less misclassification between running and walking [16]. The irregularities in gait characteristics among children with CP can also be positive for model performance, where the heterogeneity, might explain the reduced misclassification of walking in both CP groups and TD with the HAR-ChildrenCP model.

### 4.3. The effect of window size and overlapping windows

Our models confirm the significance of window settings in activity recognition, and its impact depend on the target activity. For instance, momentary activities such as jumping, are in our models best detected with the 1-second window, while cyclic activities with longer cycle lengths such as cycling is better detected with 5-seconds window. Interestingly, when the window size was increased to 5-seconds, the disparity in performance between HAR-Children and HAR-ChildrenCP models diminished in the CP Free group. This suggest that the intermittent changes in the CP Free group are concealed by majority voting in the 5 second window, and conceivably the intermittent activities in the training data play a reduced role for model performance. This data reduction effect of the larger windows have been beneficial in previous research and for specific activities [46], such as walking, cycling, and running, which have cycle rates exceeding one second. Existing literature involving children with CP typically use window sizes ranging from five to fifteen seconds [24, 33–35], reporting higher F1 Scores with larger windows [24]. Ferrari et al., [38] advocates for windows that are long enough to capture a complete cycle of a specific movement yet short enough to distinguish between similar movements.

Within the context of evaluating children's daily activities, our data suggest that the 1-second models should be used when the objective is to detect momentary activities, such as jumping. However, it is essential to recognise that these momentary actions often transpire within a broader spectrum of gross motor activities, such as running, walking, and standing, which is often of primary interest. In this context larger window sizes offer a more efficient approach to data reduction and improves model performance in typical daily life scenarios. Hence, in the context of everyday life and health outcomes the utilization of the 3-seconds model emerges as potentially preferrable. This choice is substantiated by our results, where the 3-second model

predicts momentary activities, but also activities of extended duration, including cycling, with commendable accuracy.

Regarding the choice between overlapping or non-overlapping windows, our study reveals marginal difference in model performance with slightly better accuracy observed with overlapping windows. The optimal window size represent a trade-off between processing speed and prediction accuracy [21]. Consequently, the benefits of overlapping windows must be weighed against the drawbacks, particularly in clinical application. The time-consuming nature of processing overlapping windows is a notable concern. In our model-training the 1-second with overlap required approximately ten times longer processing time than the 5-seconds without overlap. This substantial time difference is consistent with findings of Dehghani et al., [39], where segmentation with overlapping windows took twice as long, and training took four times as long compared to non-overlapping. Furthermore, in their study, the memory requirements for overlapping windows were nearly nine times greater [39]. Given the trade-off in accuracy, questions arise regarding the practical utility of overlapping windows in clinical context, especially when dealing with large data sets.

## 4.4. Strengths and limitations

Comparing HAR and ML models present considerable challenges due to variations in classifier usage, study populations, activity protocols, accelerometers, accelerometer placements, among other factors. Our study is unique for our utilization of the XGBoost classifier and two accelerometers. The use of two or more accelerometers and our placements have been emphasized as preferred settings [13, 33]. The relatively infrequent use of the XGBoost classifier in HAR research contrasts with the prevalence and recommendations in other fields, and have shown strengths, particularly due to its sequential learning [7, 13]. Moreover, our study incorporates specific configurations and specifications that have demonstrated excellence in other population groups [7, 13, 14]. Additionally, our study is strengthened and novel by the inclusion of playful behaviour and group activities, and thereby simulated habitual activities for children and children with CP. Furthermore, our data set is relatively large in comparison to other validation studies including children.

Some limitations warrant consideration. We have different group sizes and amount of training data from TD children and children with CP. Following the principle that more training data and variation in movement patterns generally benefits model training we opted to include all available data. Thus, we think that the different group sizes between TD and CP will not substantially influence the model performance. However, adding more CP training data would probably improve the model performance further, due to even more variation in the training data's movement patterns. We have limited free play training data, potentially impacting prediction accuracy for the CP Free group. Additionally, the absence of a TD Free group prevents a direct examination of the contrasting effects of free play versus the presence of CP on model performance.

## 4.5 Future perspectives and implications

The present article underscores the specificity of the HAR method, emphasizing the importance of the training data as well as technical specifications for future research and clinical work. One noteworthy consideration is the necessity for a specific HAR model for CP, GMFCS I &II as the disparity between HAR-children and HAR-ChildrenCP appear relatively minor for standardised activities. Conversely, based on our results we can suggest the call for the HAR-ChildrenCP model, or children with disabilities model, is not restricted to children with CP, but also to TD children. Such a model, like our HAR-ChildrenCP, may

offer enhanced versatility by greater variation within the training data, facilitating the recognition of a broader spectrum of movement patterns in the general child population. An intriguing avenue for future research is to expand the free play data set, which involves training our model with activity data collected in children's daily environment, such as in their home, kindergarten, school, and leisure activities. This further expansion into real-world scenarios could yield valuable insights and advance the applicability of our model and HAR methods in child populations. Furthermore, our development of a HAR-model, and future enhancements of such a model for children with CP, enables the examination of treatment effects of habilitation interventions that aims to influence the amount, patterns, or distribution of PA. A potential limitation for future clinical use is sensor misplacement, as it is crucial to ensure correct alignment with gravity for accurate data collection. Therefore, it's important to educate participants on proper sensor placement, especially when positioning sensors outside the clinic.

The accuracy of our sub-models is between 75–95%, which is comparable with previous studies. As described above our model demonstrates greater performance in certain activities and incorporates free play and childlike behavior. However, there remains a debate if our results are good enough, as we recognize some challenges regarding activities during free play. Therefore, in addition to expanding the free play data set, it may be beneficial to explore alternative approaches, such as deep learning experiments. In fact, to optimize activity recognition for future use, it would be intriguing to compare various HAR methods for children, particularly for those with deviating movement patterns.

## 5.Conclusion

Both HAR models demonstrate precise predictions for standardised activities in both TD children and those with CP, and slightly less precise predictions for free play activities, but still precises and favourable compared to previous models. Among the three groups, the CP Stan has the most accurate predictions, prompting consideration of the influence of impairment versus free play activities. Based on the highest overall accuracy, the NTNU HAR-ChildrenCP model with 1-second window would be recommended for all three test groups. However, the optimal window size and overlap depend on the target activity. For activities such as cycling, 3- or 5-seconds windows perform better. Considering the ability to predict both momentary activities and activities of extended duration, the 3-second window without overlap would be recommended for population measurements.

## Supporting information

**S1 Table. Overview of conducted activities for each test group.**
(TIF)

**S2 Table. Definition of activities.**
(TIF)

**S3 Table. Overall accuracy for each model with and without overlap and original activity classes.** Test groups: TD = typically developing children, CP Stan = Cerebral palsy with standardised activities, CP Free = Cerebral palsy with Free play. The range of accuracy is 0–1, where higher scores are better.
(TIF)

**S1 Fig. Overview of the 12 sub-models.** TD children = typically developing children, CP = Cerebral Palsy, O = Overlap, Test groups: TD = typically developing children, CP

Stan = Cerebral palsy with standardised activities, CP Free = Cerebral palsy with Free play.
(TIF)

## Acknowledgments

We thank the children and families that participated in this study, and other contributors that assisted with the data collection especially Roar M. Fenne. We also thank Kerstin Bach for the utilization of the NTNU-HAR method.

## Author Contributions

**Conceptualization:** Siri Merete Brændvik, Karin Roeleveld, Ellen Marie Bardal.

**Data curation:** Marte Fossflaten Tørring, Aleksej Logacjov, Astrid Ustad, Karin Roeleveld, Ellen Marie Bardal.

**Formal analysis:** Marte Fossflaten Tørring, Karin Roeleveld.

**Funding acquisition:** Siri Merete Brændvik, Karin Roeleveld, Ellen Marie Bardal.

**Methodology:** Marte Fossflaten Tørring, Aleksej Logacjov, Astrid Ustad, Ellen Marie Bardal.

**Project administration:** Siri Merete Brændvik, Karin Roeleveld, Ellen Marie Bardal.

**Software:** Marte Fossflaten Tørring, Aleksej Logacjov.

**Supervision:** Siri Merete Brændvik, Karin Roeleveld.

**Visualization:** Marte Fossflaten Tørring.

**Writing – original draft:** Marte Fossflaten Tørring.

**Writing – review & editing:** Marte Fossflaten Tørring, Aleksej Logacjov, Siri Merete Brændvik, Astrid Ustad, Karin Roeleveld, Ellen Marie Bardal.

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
