## [Decision Letter · Decision Letter 0]

8 Jan 2024

PONE-D-23-37915Validation of two novel human activity recognition models for typically developing children and children with Cerebral Palsy.PLOS ONE

Dear Dr. Tørring,

Thank you for submitting your manuscript to PLOS ONE. After careful consideration, we feel that it has merit but does not fully meet PLOS ONE’s publication criteria as it currently stands. Therefore, we invite you to submit a revised version of the manuscript that addresses the points raised during the review process.

We look forward to receiving your revised manuscript.

Kind regards,

Jyotindra Narayan

Academic Editor

PLOS ONE

Journal Requirements:

6. We notice that your supplementary figures are uploaded with the file type 'Figure'. Please amend the file type to 'Supporting Information'. Please ensure that each Supporting Information file has a legend listed in the manuscript after the references list.

Additional Editor Comments:

The reviewers have praised the work for significant contributions on human activity recognition in typically developing children and those with Cerebral Palsy but suggests improvement in addressing methodological concerns. Specific suggestions include clarifying data acquisition methods, questioning biases in training models (by both reviewer 1 and 3), and addressing discrepancies in group sizes. The reviewers 2 and 3 have common concerns about tangential literature and references used. The reviewer 1 have marked suggestions over the manuscript's pdf (see attachment), helping authors to address minor technical and langauge concerns. Finally, per the reviewer 3 suggestions, the authors are urged to discuss the implications of group sizes and consider ways to enlarge the free play dataset.

Reviewers' comments:

Reviewer's Responses to Questions

**Comments to the Author**

1. Is the manuscript technically sound, and do the data support the conclusions?

Reviewer #1: Yes

Reviewer #2: Yes

Reviewer #3: Yes

2. Has the statistical analysis been performed appropriately and rigorously? 

Reviewer #1: Yes

Reviewer #2: Yes

Reviewer #3: Yes

3. Have the authors made all data underlying the findings in their manuscript fully available?

Reviewer #1: No

Reviewer #2: Yes

Reviewer #3: Yes

4. Is the manuscript presented in an intelligible fashion and written in standard English?

Reviewer #1: Yes

Reviewer #2: Yes

Reviewer #3: Yes

5. Review Comments to the Author

Reviewer #1: typically developed children. Overall, on the technical side, the presented manuscript is technically sound, and all the result data support the authors' claims in the result and conclusion sections. The authors' choice of F-1, understanding of F-1, and usage are respectable. Under the microscope, however, there are minor editorial/technical errors—for example, the authors' inconsistent citation style or difficulty in explaining their 12 models concept. Furthermore, the authors haven't made any data available for public consumption. Please refer to the reviewer attachment for a complete list of questions/comments in the form of notes embedded in the manuscript PDF. Overall, the reviewer concluded that this manuscript is legible and technically sound, and all the analyses have been done appropriately.

Reviewer #2: Authors have presented an ML-based Human Activity Recognition (HAR) model for typically developing (TD) children and children with Cerebral Palsy (CP).

The study discusses the advantages of HAR models for the same and challenges in the setup and hyperparameters such as training data, optimal accelerometer placements,

and window size considerations. It presents the usage of the XGBoost classifier (12 sub-models) with 1, 3, and 5 sec time windows. The ML-based approach [1, 2, 3] is a very common HAR.

Following are my comments to further improve this article:

1. The applicability of the study may be broadened by including the usage of HAR in different health and rehabilitation fields such as exercise/YOGA pose classification [4], rehabilitation of individuals after sports injury, and improving the posture for the different exercises and training.

2. The study can also include a) a comparison of deep learning-based models [3, 4, 6] along with ML-based models for the HAR model for TD and CP children and b) a study of multimodal features [6] in HAR. Authors may refer to the studies [1, 5, 6] for choosing appropriate ML and DL methods for the task.

3. The author should consider the experimentation of multiple related datasets such as WISDM Activity Prediction, UCI HAR, DSADS, etc. Please refer to Table 1 from [6] for a list of publicly available datasets of HAR.

4. The possible direction to measure the automatic improvement in CP children after the rehab/therapy would further boost the applicability of research in a real-world application.

5. In line 73 of the Introduction section, TD is mistyped as TP.

6. In line 243 of section 2.6, it is written as F-measure is the harmonic mean between precision and sensitivity, where as it is the harmonic mean between precision and recall [7].

References:

1. Gupta, N., Gupta, S.K., Pathak, R.K. et al. Human activity recognition in artificial intelligence framework: a narrative review. Artif Intell Rev 55, 4755–4808 (2022). https://doi.org/10.1007/s10462-021-10116-x

2. Csizmadia, G., Liszkai-Peres, K., Ferdinandy, B. et al. Human activity recognition of children with wearable devices using LightGBM machine learning. Sci Rep 12, 5472 (2022). https://doi.org/10.1038/s41598-022-09521-1

3. D. Sakkos, K. D. Mccay, C. Marcroft, N. D. Embleton, S. Chattopadhyay and E. S. L. Ho, "Identification of Abnormal Movements in Infants: A Deep Neural Network for Body Part-Based Prediction of Cerebral Palsy," in IEEE Access, vol. 9, pp. 94281-94292, 2021, doi: 10.1109/ACCESS.2021.3093469.

4. Vallabhaneni, N., Prabhavathy, P. Segmentation quality assessment network-based object detection and optimized CNN with transfer learning for yoga pose classification for health care. Soft Comput (2023). https://doi.org/10.1007/s00500-023-08863-w

5. Wan, S., Qi, L., Xu, X. et al. Deep Learning Models for Real-time Human Activity Recognition with Smartphones. Mobile Netw Appl 25, 743–755 (2020). https://doi.org/10.1007/s11036-019-01445-x

6. Kaixuan Chen, Dalin Zhang, Lina Yao, Bin Guo, Zhiwen Yu, and Yunhao Liu. 2021. Deep Learning for Sensor-based Human Activity Recognition: Overview, Challenges, and Opportunities. ACM Comput. Surv. 54, 4, Article 77 (May 2022), 40 pages. https://doi.org/10.1145/3447744

7. Schütze, H., Manning, C. D., & Raghavan, P. (2008). Introduction to information retrieval (Vol. 39, pp. 234-265). Cambridge: Cambridge University Press.

Reviewer #3: The article “Validation of two novel human activity recognition models for typically developing children and children with Cerebral Palsy” is well written and pleasurable to read. It presents a classifying method of physical activities in CP children and above all the influence of different parameters of this method: learning dataset composed of only TD children or a mix of TD and CP children, the length of the observation window, the choice to use or not an overlap to define this window. Results are clear and complete which helps to understand the effect of each parameter. Collecting and processing the dataset has been a very consequent job that might be very useful for any research team working on physical activity recognition in CP children in real life which is a crucial question. I would like to sincerely thank the authors for that huge and useful work.

Nevertheless, some methodological choices and the ensuing elements of discussion should be address to strengthen the impact of this study.

Specific comments

L44: As soon as the abstract you should indicate how data are acquired (video, IMU, … ?)

L85: It is a bit surprising that in most of previous studies, the authors did not think that training the model with one population (adults) would be a bias when using the model with another one (children). This bias is presented as on major argument for your study. However, I am pretty sure this potential bias is largely exaggerated as in most studies aiming at recognize human activities the training population has the same characteristics as the target one.

L99: And even, the three articles you present come from the same research team in Queensland, Australia. I am very surprised that only two research teams in the world are interested in PA recognition in CP. Are you only focusing on the articles in which the methodology used for this recognition is very close to yours so that comparison is easier? If so that needs to be modified. You should be able to compare your results even when the classifier’s principle is different from yours.

L115: It is not clear what are the ‘two’ models. There is no prior justification for the need of two different models and what will be the advantage of one vs. the other, or in which cases one should be used. Just after, I understand that you mean two models because learning data are not the same, e.g. one with only TD and the other one mixing TD and CP. The question is to know if is the methodology identical in both cases. If it is, I would rather say that there is actually only one model with which you are testing the influence of identifying the model’s parameters from two learning datasets. Indeed, afterwards you are talking about 12 models when combining this learning dataset parameter to two other ones, i.e. length of window (1, 3 or 5 s) and the overlap.

L123: The two groups have very different size (ratio 1:4). It means that in your second dataset, mixing TD and CP, TD activities are actually the large majority. On L201, you don not indicate if the second dataset includes all data from TD children. I thus assume that it is the case. It actually means that your second ‘model’ center of gravity is actually drifted towards TD movements.

L155: What is the sensitivity of your method to accelerometer misplacement? The final goal is to monitor activities in real life situation in which CP children or their parents will place the devices. They are not clinicians trained to the use of such device. One usual error in that case is axes misalignment. It is important to test this robustness to that kind of bias. In your experiment, these biases were absent as all the devices were placed by the team (maybe even the same operator). It implies that the data used in the LOSOCV evaluation is homogeneous regarding these biases. As being outside the learning database, the child left-out represents a subject in real life. But actually, sensors’ placement on a child in real life will be less precise.

L197: Fig. 1 is hardly readable.

Discussion: you should discuss a bit more about the groups’ sizes. I agree with you that your whole dataset with 79 children is quite large compared to other studies. But 80% of them are TD. And moreover, as some activities (e.g. jumping) is more difficult for CP children, when focusing more finely on the dataset activity per activity, this ratio is sometimes even more in favor of TD children.

L462: I agree with you that it is always a bit frustrating not to have more free play datasets. Free play, even in clinical environment, is one important way to be closer to real life. You honestly recognize that your free play dataset should be enlarged but not indicating how this could be done. If you have an idea, could you explain it briefly?

6. PLOS authors have the option to publish the peer review history of their article (what does this mean?). If published, this will include your full peer review and any attached files.

Reviewer #1: No

Reviewer #2: **Yes: **Dr. Durgesh Kumar

Reviewer #3: No

---

## [Author Response · Author response to Decision Letter 0]

21 Feb 2024

Response to editor and reviewers. 

Our response are provided under the comments.

In the word file (response to editor and reviewers) the response are in blue, page and line numbers refer to the manuscript file with track changes. 

Journal Requirements:

Response: Thank you for this, we have checked the guidelines again and made minor changes. 

We have changed the file names of our figures after the standard. We have also changed our referral to these in the manuscript according to the guidelines. We have also changed the frontpage according to the guidelines, we have removed the authors contribution, page 2, as it seems to only be required in the submission form.

Response: Thank you for pointing this out. We have updated our manuscript and the ethics statement in the submission form. Page 7, line 137-141: “All participants were given written age-appropriate information about the study. This was distributed at school and taken home together with written information and consent forms to their parents/guardians. If the parents gave their written consent, verbal information was given to the children before the study started”.

Response: Our code is a further development of previous code from our research group at NTNU. This code is already uploaded to GITHUB (https://github.com/ntnu-ai-lab/harth-ml-experiments). Our code includes minor changes to this code to fit the scope of our study. Up on publication our code will be provided as well. 

4. We note that the grant information you provided in the ‘Funding Information’ and ‘Financial Disclosure’ sections do not match. When you resubmit, please ensure that you provide the correct grant numbers for the awards you received for your study in the ‘Funding Information’ section.

Response: We have corrected our funding information in the submission form. We do not have a specific grant number for this work. We have also updated our cover letter to include information of financial disclosure. 

Response: We have created a platform using Dataverse (https://dataverse.no/) where all our information regarding the data set will be published. Currently the cite is registered without the raw datafiles. However, all other metadata is provided, and the files will be uploaded on acceptance. Here is the DOI number (https://doi.org/10.18710/EPCXCC) that we refer to in our data availability statement. The DOI will be activated up on acceptance. 

6. We notice that your supplementary figures are uploaded with the file type 'Figure'. Please amend the file type to 'Supporting Information'. Please ensure that each Supporting Information file has a legend listed in the manuscript after the references list.

Response: We are sorry for the mislabeling. We have changed the file type and file name according to guidelines. 

Additional Editor Comments:

The reviewers have praised the work for significant contributions on human activity recognition in typically developing children and those with Cerebral Palsy but suggests improvement in addressing methodological concerns. Specific suggestions include clarifying data acquisition methods, questioning biases in training models (by both reviewer 1 and 3), and addressing discrepancies in group sizes. The reviewers 2 and 3 have common concerns about tangential literature and references used. The reviewer 1 have marked suggestions over the manuscript's pdf (see attachment), helping authors to address minor technical and langauge concerns. Finally, per the reviewer 3 suggestions, the authors are urged to discuss the implications of group sizes and consider ways to enlarge the free play dataset.

Response: We are grateful for the thoughtful comments and suggestions from the reviewers and the editorial office, which allowed us to improve our manuscript significantly. We have addressed the comments about tangential literature by providing additional references in our manuscript. We detail below how we responded to each concern from the reviewers point-by-point. 

Reviewers' comments:

Reviewer's Responses to Questions

Comments to the Author

1. Is the manuscript technically sound, and do the data support the conclusions?

Reviewer #1: Yes

Reviewer #2: Yes

Reviewer #3: Yes

2. Has the statistical analysis been performed appropriately and rigorously?

Reviewer #1: Yes

Reviewer #2: Yes

Reviewer #3: Yes

3. Have the authors made all data underlying the findings in their manuscript fully available?

Reviewer #1: No

Reviewer #2: Yes

Reviewer #3: Yes

4. Is the manuscript presented in an intelligible fashion and written in standard English?

Reviewer #1: Yes

Reviewer #2: Yes

Reviewer #3: Yes

5. Review Comments to the Author

Reviewer #1:

Typically developed children. Overall, on the technical side, the presented manuscript is technically sound, and all the result data support the authors' claims in the result and conclusion sections. The authors' choice of F-1, understanding of F-1, and usage are respectable. Under the microscope, however, there are minor editorial/technical errors—for example, the authors' inconsistent citation style or difficulty in explaining their 12 models concept. Furthermore, the authors haven't made any data available for public consumption. Please refer to the reviewer attachment for a complete list of questions/comments in the form of notes embedded in the manuscript PDF. Overall, the reviewer concluded that this manuscript is legible and technically sound, and all the analyses have been done appropriately.

Response: 

Thank you for your comments and feedback. We have made changes accordingly. 

- Regarding the F1 Score, see point 9 in the specific comment section bellow. 

- Our citations and references are corrected according to Vancouver reference style. 

- We have made several changes to the document regarding the 12 sub-model explanation, which will make it easier for the reader to follow our argumentation. See new supporting figure 4 (S4 Fig) and revised manuscript on page 10, line 222-224. “The two main models comprise a total of 12 sub-models, 6 for each main model, covering 1-, 3-, and 5-seconds windows with and without overlapping windows, see supporting information (S4 Fig) for overview of the 12 sub-model creation”. 

- Regarding the data availability we have created a cite on Dataverse where the data will be published upon exception/publication (https://doi.org/10.18710/EPCXCC). Our code can be found with similar codes from our research team at GITHUB: https://github.com/ntnu-ai-lab/harth-ml-experiments. 

Comments from the manuscript file: 

1. Line 79. Missing definition.

Response: Thank you for this comment. Definition of TD is typically developing, we have made a typo on page 4 line 76. We have now corrected the typo and the definition is there. 

2. Line 80. This notion is in conflict with previous statement.

Response: We agree that the sentences needed adjustments. We have rewritten the paragraph. 

Page 4/5, line 81-86. 

3. Line 89. Citations?

Response: We have rewritten the sentence and added references for the updated prevalence. Page 5, line 96-98. “CP is one of the most frequent disability among children (25) with the global prevalence of approximately 1.6 /1000 live births in high income countries (26).” 

4. Line 99. Wrong citation style.

Response: We have corrected the format. 

5. Line 172-174. Was the labelling done with one person or more? If more, an addition of Cohen's kappa agreement coefficient would be beneficial in justifying the dataset’s quality by looking at the reliability between the rater(labeller).

Response: More than one person labelled the data. In Ustad et al., (2023) and Bach et al., (2022) they have reported interrater reliability, with the coefficient of >0.95. Using the same method and activity definitions and overlapping raters as our study. We have added this information in our manuscript on page 9, line 189-191. 

Ustad A, Logacjov A, Trollebø SØ, Thingstad P, Vereijken B, Bach K, et al. Validation of an Activity Type Recognition Model Classifying Daily Physical Behavior in Older Adults: The HAR70+ Model. Sensors. 2023;23(5):2368.

Bach K, Kongsvold A, Bårdstu H, Bardal EM, Kjærnli HS, Herland S, et al. A Machine Learning Classifier for Detection of Physical Activity Types and Postures During Free-Living. Journal for the Measurement of Physical Behaviour. 2022;1(aop):1-8.

6. Line 183. Why 50% overlap? 

Response: When employing HAR models with fixed windows there is a potential loss of some activities at the endpoint of the window. The choice of a 50% overlap was deliberate to ensure that no single endpoint in the data was overlooked as the new window would be in the middle of the previous window. The 50% overlap is commonly used in HAR, when employing these fixed windows. We considered increasing the overlap, however, as it would substantially increase the processing time it will make the model unsuitable for clinical purposes. Under we have provided two references that also use 50% overlap in their HAR models. We have added a sentence on this in our manuscript on page 9, line 199-200. “The overlap was included to investigate potential loss of activities at the endpoint of the window”.

Lara OD, Pérez AJ, Labrador MA, Posada JD. Centinela: A human activity recognition system based on acceleration and vital sign data. Pervasive and mobile computing. 2012 Oct 1;8(5):717-29.

Ferrari A, Micucci D, Mobilio M, Napoletano P. On the personalization of classification models for human activity recognition. IEEE Access. 2020 Feb 12;8:32066-79.

7. Line 191. Pleas don’t forget to cite the original XGBoost paper: https://arxiv.org/abs/1603.02754

Response: Thank you for providing this reference for us. We have cited at page 10, line 208. 

8. Line 201-202. A table or a more descriptive definition, whether in the text or the supplement section, would help greatly digest this sentence.

Response: We agree and have provided a figure in the supplementary information (S4 Fig) that explain how we get to 12 sub models and how these are linked. We have revised our manuscript page 10, line 217-224. 

9. Line 204 and 205. Why use k-fold here and LOSOCV in the other section? A K-fold CV might have higher accuracy but is more likely to be artificial due to intra-subject contamination, unlike LOSOCV. Additionally, leave-n-subject-out cross-validations might be more beneficial as they provide a cross-subject generalisability (subject-independent performance) overview while maintaining a manageable number of cross-validation-round. 

Response: Regarding the use of K-fold or LOSOCV we have only used the method of LOSOCV. We have used the LOSOCV method as a k-fold on 6 subjects, or as you suggested leave-n-subjects-out-cross-validation. We have revised the manuscript, and we are sorry for the confusion. Modifications on page 10, line 224-225. “For each model we initially performed a leave-six-subjects-out-cross-validation with hyperparameter optimization in the form of a grid search”. 

10. Line 239. Are they class weighted or unweighted?

Response: We have calculated macro/ non-weighted F1 Score using the following formula F1 Score= 2x precision x recall / precision + recall. As we only report class-wise F1 Scores we have concluded that there is no need for calculating the weighted F1 Score in our results. Updated manuscript on page 12, line 265. 

Reviewer #2: 

Authors have presented an ML-based Human Activity Recognition (HAR) model for typically developing (TD) children and children with Cerebral Palsy (CP).

The study discusses the advantages of HAR models for the same and challenges in the setup and hyperparameters such as training data, optimal accelerometer placements,

and window size considerations. It presents the usage of the XGBoost classifier (12 sub-models) with 1, 3, and 5 sec time windows. The ML-based approach [1, 2, 3] is a very common HAR. 

Response: Thank you for your comments on our manuscript and suggestions for further work and improving the usability of human activity recognition models. Additionally, we would thank you for

---

## [Decision Letter · Decision Letter 1]

1 Apr 2024

PONE-D-23-37915R1Validation of two novel human activity recognition models for typically developing children and children with Cerebral Palsy.PLOS ONE

Dear Dr. Tørring,

Thank you for submitting your manuscript to PLOS ONE. After careful consideration, we feel that it has merit but does not fully meet PLOS ONE’s publication criteria as it currently stands. Therefore, we invite you to submit a revised version of the manuscript that addresses the points raised during the review process.

We look forward to receiving your revised manuscript.

Kind regards,

Jyotindra Narayan

Academic Editor

PLOS ONE

**Additional Editor Comments:**

Reviewer #1 recommends accepting the manuscript, acknowledging its contribution to Human Activity Recognition (HAR) in typically developing children and children with cerebral palsy. While noting improvements in presentation and writing, Reviewer #2 raises concerns about the lack of novelty in methodology and research objectives. Suggestions for enhancement include incorporating a deep learning-based model for comparison with XGBoost, utilizing multimodal features combining video image and sensor data, sharing data with the research community, and addressing how the proposed models differ from existing methods. Reviewer #3 appreciates the authors' responses to concerns but suggests including unpublished results regarding sensor misplacement's impact on model performance as a potential limitation.

Reviewers' comments:

Reviewer's Responses to Questions

**Comments to the Author**

1. If the authors have adequately addressed your comments raised in a previous round of review and you feel that this manuscript is now acceptable for publication, you may indicate that here to bypass the “Comments to the Author” section, enter your conflict of interest statement in the “Confidential to Editor” section, and submit your "Accept" recommendation.

Reviewer #1: All comments have been addressed

Reviewer #2: (No Response)

Reviewer #3: (No Response)

2. Is the manuscript technically sound, and do the data support the conclusions?

Reviewer #1: Yes

Reviewer #2: Partly

Reviewer #3: (No Response)

3. Has the statistical analysis been performed appropriately and rigorously? 

Reviewer #1: Yes

Reviewer #2: Yes

Reviewer #3: (No Response)

4. Have the authors made all data underlying the findings in their manuscript fully available?

Reviewer #1: Yes

Reviewer #2: No

Reviewer #3: (No Response)

5. Is the manuscript presented in an intelligible fashion and written in standard English?

Reviewer #1: Yes

Reviewer #2: Yes

Reviewer #3: (No Response)

6. Review Comments to the Author

Reviewer #1: (No Response)

Reviewer #2: The manuscript solves a real-world problem of Human Activity Recognition (HAR) in normal children and children with celebration palsy. The authors have improved in presentation and writing from the first revised version. However, the manuscript lacks novelty in terms of the proposed methodology and research Objectives. The significant research contributions of the study are as follows:

1. The novel datasets for Human Activity Recognition for Typically Developing Children (TD) and Children with Cerebral Activity.

2. Experiment and Analysis of HAR on the above datasets using XGBoost classifier.

3. Study of the impact of windows size (1 sec, 3 sec, 5 sec) and with overlapping.

In my first review, I have provided possible direction to extend the novelty and technical contribution of the paper by including:

1. Inclusion of a Deep learning-based model (such LSTM) for comparison with XGBoost: The author refused to experiment with the model, citing their assumption that it would not improve their results further without even experimenting with it. LSTM and BiLSTM are known to improve the results in sequential data. Further, the authors claimed there was a lack of space to fit their results, as they had too many large result tables. The results table could appropriately be compressed by representing it with suitable diagrams.

2. Inclusion of multimodal features: I mean to use both the video image and sensor data for the HAR. As the confusion matrix (Figure 5) NNTU HAR ChildrenCP-5 sec shows, some of the Walking and running data is wrongly predicted as Standing. In such a scenario, the image taken from the camera could assist the sensor data in proper activity recognition.

3. The authors have highlighted the lack of publicly available data for HAR in children and children with CP. Unless researchers share their data for academic and research purposes, this problem will remain as it is. Therefore, it is requested that the authors share their data with the research community after the journal accepts it. Please share the data sharing policy and some data samples with the Editorial Team before the publication of the papers.

4. How much do the Model NNTU-HAR-Children and NNTU-HAR-ChildrenCP differ from the NNTU-HAR method proposed by Roar M. Fenne?

Other Minor comments:

1. In section 2.4.2 (Video Annotation), it is not clear how many annotators were there in total, and each video frame annotated by how many annotators, and what was the score inter-annotator agreement?

2. The research has not concluded the optimal time window for the HAR in their conclusion section and results discussion section.

Reviewer #3: The authors provided convincing answers to my concerns and consequently modified the article. There is only one concern for which extra information would be useful. Regarding robustness to sensors' misplacement, the authors explained that "Our research group have tested the consequence of sensor misplacement on model performance in previous studies using

the same methodology (unpublished). As long as the sensors are placed in the correct direction towards gravity minor deviations in placements does not affect the accuracy of the model." Even if unpublished yet, the results of these previous studies could be presented in discussion as a potential limitation already taken into account. In the current version, it is as if the limitation does not exist.

7. PLOS authors have the option to publish the peer review history of their article (what does this mean?). If published, this will include your full peer review and any attached files.

Reviewer #1: No

Reviewer #2: **Yes: **Dr. Durgesh Kumar

Reviewer #3: No

---

## [Author Response · Author response to Decision Letter 1]

9 May 2024

Additional Editor Comments:

Reviewer #1 recommends accepting the manuscript, acknowledging its contribution to Human Activity Recognition (HAR) in typically developing children and children with cerebral palsy. While noting improvements in presentation and writing, Reviewer #2 raises concerns about the lack of novelty in methodology and research objectives. Suggestions for enhancement include incorporating a deep learning-based model for comparison with XGBoost, utilizing multimodal features combining video image and sensor data, sharing data with the research community, and addressing how the proposed models differ from existing methods. Reviewer #3 appreciates the authors' responses to concerns but suggests including unpublished results regarding sensor misplacement's impact on model performance as a potential limitation.

Response to editor: 

Dear Jyotindra Narayan 

We would like to express our gratitude for the thorough review process and the insightful comments provided by the reviewers and the editorial office on our manuscript. We appreciate the time and effort invested by all involved, which has undoubtedly contributed to the improvement of our work. We have carefully considered each comment and suggestion raised by the reviewers and have made necessary revisions to our manuscript accordingly. 

Regarding the suggestions put forth by Reviewer #2, we respectfully acknowledge the reviewer's perspective and the importance of exploring various avenues for advancement in Human Activity Recognition (HAR). However, we must clarify that our primary objective is to develop and validate a HAR model tailored to typically developing children and children with cerebral palsy, with deviating movement pattern, by using existing HAR methodology. Our objective was not to compare two methodologies for activity recognition but validate two models. 

Reviewer 2 has raised a valid point regarding the novelty of our study's method and objectives. We acknowledge that our study does not introduce significant novelty in terms of methodology, as there already exist Human Activity Recognition (HAR) models employing various machine learning techniques. However, it is worth noting that few of these models are built on data from children, and only three focus on children with cerebral palsy (CP). From our perspective novelty in our study is evident in several aspects. Firstly, the uniqueness of our data set stands out, as it includes a substantial number of typically developing (TD) children alongside children with CP. This diverse data set provides a rich foundation for our analysis. Secondly, the inclusion of free play activities, including childlike activities and ball games, adds a novel dimension. These activities reflect real-world scenarios and add depth to our understanding of activity recognition in children. Thirdly, our comparison between TD and CP models is unprecedented, particularly testing a model designed for children with deviating movement patterns on typically developing children. Lastly, our exploration of window sizes and overlap also contributes valuable insights to the field, by offering insights into optimal parameter settings for activity recognition systems.

In addition to our points raised in the discussion regarding future perspectives and implications, we acknowledge the need for further improvements to HAR methodology. In response to Reviewer 2's suggestion, we have included a new paragraph under this heading about deep learning as a potential avenue for future optimization of HAR in general. We fully recognize the value of further exploration of different methodologies as deep learning, but we believe they warrant dedicated studies of their own. Therefore, we kindly request your understanding in accepting our manuscript without additional experiments that incorporating multimodal features and deep learning-based models.

Once again, we extend our sincere appreciation to the reviewers and the editorial office for their valuable feedback and consideration of our work. We remain committed to contributing to the advancement of knowledge in the field of Human Activity Recognition and look forward to the opportunity to share our findings with the research community.

Thank you for your attention to this matter.

Yours sincerely,

Marte Fossflaten Tørring 

NTNU

Review Comments to the Author

Reviewer #1:

(No Response)

Reviewer #2: 

The manuscript solves a real-world problem of Human Activity Recognition (HAR) in normal children and children with celebration palsy. The authors have improved in presentation and writing from the first revised version. However, the manuscript lacks novelty in terms of the proposed methodology and research Objectives. The significant research contributions of the study are as follows:

1. The novel datasets for Human Activity Recognition for Typically Developing Children (TD) and Children with Cerebral Activity.

2. Experiment and Analysis of HAR on the above datasets using XGBoost classifier.

3. Study of the impact of windows size (1 sec, 3 sec, 5 sec) and with overlapping.

In my first review, I have provided possible direction to extend the novelty and technical contribution of the paper by including:

1. Inclusion of a Deep learning-based model (such LSTM) for comparison with XGBoost: The author refused to experiment with the model, citing their assumption that it would not improve their results further without even experimenting with it. LSTM and BiLSTM are known to improve the results in sequential data. 

Response:

Thank you for the time and effort you spent on our manuscript. We appreciate your thoughtful consideration of our work and your suggestions for improvements. We acknowledge that our study does not introduce significant novelty in terms of technical methodology, as there already exist Human Activity Recognition (HAR) models employing various machine learning techniques. However, it is worth noting that few of these models are built on data from children, and even fewer focus on children with cerebral palsy (CP).

From our perspective novelty in our study is evident in several aspects. Firstly, the uniqueness of our data set stands out, as it includes a substantial number of typically developing (TD) children alongside children with CP. This diverse data set provides a rich foundation. Secondly, the inclusion of free play activities, including childlike activities and ball games, adds a novel dimension. These activities reflect real-world scenarios and add depth to our understanding of activity recognition in children. Thirdly, our comparison between a TD and CP model is unprecedented, particularly testing a model designed for children with deviating movement patterns on typically developing children. Lastly, our exploration of window sizes and overlap also contributes to valuable insights to the field, by offering insights into optimal parameter settings for activity recognition systems. Our focus stems from our commitment to addressing clinical needs and facilitating activity identification in children with CP. We have clarified this further in the manuscript.

We apologize for giving the impression that we didn't think the suggested models would have any potential improving activity detection. We acknowledge that including additional experiments, as you suggest, could potentially yield some improved results and novelty in terms of methodology. We have included a new paragraph under the heading future perspectives and implications about deep learning as a potential avenue for future optimization of HAR in general (page 25, line 485-492). We fully recognize the value of further exploration of different methodologies as deep learning, but we believe they warrant dedicated studies of their own. Therefore, in our paper, we adhere to the original scope, objectives, and methods of our study. We have clarified our objective on page 5, line 95-102. We hope this clarification resonates with your understanding of our research goals and constraints.

Further, the authors claimed there was a lack of space to fit their results, as they had too many large result tables. The results table could appropriately be compressed by representing it with suitable diagrams.

Response: 

We apologize for the words chosen in our previous answer and we thank you for the suggested solution. However, our decision not to expand the experiments beyond the scope of our current study was based on several factors, including the choice to maintain focus on our research question within the clinical context. Introducing further experiments is not solely restricted to space limitations but also risks complicating the manuscript for readers, particularly those outside the technological community.

 Moreover, we would like to refer to our answer to your first remark; that expanding the scope of experiments would necessitate the establishment of new objectives. Given our commitment to ensuring accessibility and understanding, especially among clinicians and other stakeholders in the clinical community, we must prioritize clarity and conciseness in our presentation.

2. Inclusion of multimodal features: I mean to use both the video image and sensor data for the HAR. As the confusion matrix (Figure 5) NNTU HAR ChildrenCP-5 sec shows, some of the Walking and running data is wrongly predicted as Standing. In such a scenario, the image taken from the camera could assist the sensor data in proper activity recognition.

Response:

It seems we may have misunderstood each other, and we apologize for any confusion. In our study, we aim to further develop and validate the use of small wearable sensor technology to monitor activity in everyday life of children with cerebral palsy. In our study, the video recordings are used as a “gold standard” which the model is tested against. If we understand correctly, you're proposing the utilization of video recordings from the children to enhance the model's performance. While integrating multimodal features as you suggest could undoubtedly optimize the activity recognition model, we must consider the practical constraints within our clinical context. Incorporating 24/7 video recordings from children's everyday lives would present several challenges, including privacy concerns, ethical considerations, and increased burden on the patients and is therefore not feasible. 

3. The authors have highlighted the lack of publicly available data for HAR in children and children with CP. Unless researchers share their data for academic and research purposes, this problem will remain as it is. Therefore, it is requested that the authors share their data with the research community after the journal accepts it. Please share the data sharing policy and some data samples with the Editorial Team before the publication of the papers.

Response:

We totally agree with the importance of data sharing in advancing research efforts. In response to this concern, we have developed a comprehensive data sharing plan, which we previously shared with the editorial staff and included in our response to reviewer 1 during the previous review round. Here is a recap of our data sharing plan:

Our data will be made publicly available upon acceptance/publication through a dedicated repository on Dataverse, accessible via the following link: https://doi.org/10.18710/EPCXCC. Additionally, we have ensured that our codebase is readily accessible for fellow researchers and interested parties. You can find our code, along with similar codes from our research team, on our GitHub repository: https://github.com/ntnu-ai-lab/harth-ml-experiments.

We are committed to fostering collaboration and contributing to the collective advancement of knowledge in this field. We appreciate your attention to this matter and your dedication to promoting open science principles. We will share data with the editorial office upon request from the editor. 

4. How much do the Model NNTU-HAR-Children and NNTU-HAR-ChildrenCP differ from the NNTU-HAR method proposed by Roar M. Fenne?

Response:

Our models, NTNU-HAR-Children and NTNU-HAR-ChildrenCP, are developed using training data from children and adolescents, and are tailored to recognized activity patterns often seen in children. The master’s thesis by Roar M. Fenne tested how well a model trained with data from adults performed in recognizing activities in children. For more information about the adult model used in the study by Fenne, I recommend exploring the recent publication titled "A Machine Learning Classifier for Detection of Physical Activity Types and Postures During Free-Living" by Bach et al. 2022. 

Other Minor comments:

1. In section 2.4.2 (Video Annotation), it is not clear how many annotators were there in total, and each video frame annotated by how many annotators, and what was the score inter-annotator agreement?

Response: 

Thank you for noting this unclarity in section 2.4.2 regarding the video annotation process. We have considered your suggestion and updated our manuscript accordingly. On page 8, lines 161-165 now provide more clarity on the annotation process:

“After clarifying the activity definitions and discussions based on video examples four raters annotated each video independently. Inter-rater reliability of > 0.95 on this methodology has been reported in earlier studies using the same methods, activity definitions and overlapping raters with the present study (Bach et al., 2022; Ustad et al., 2023).”

Bach, K., Kongsvold, A., Bårdstu, H., Bardal, E. M., Kjærnli, H. S., Herland, S., Logacjov, A., & Mork, P. J. (2022). A Machine Learning Classifier for Detection of Physical Activity Types and Postures During Free-Living. Journal for the Measurement of Physical Behaviour, 1(aop), 1-8. 

Ustad, A., Logacjov, A., Trollebø, S. Ø., Thingstad, P., Vereijken, B., Bach, K., & Maroni, N. S. (2023). Validation of an Activity Type Recognition Model Classifying Daily Physical Behavior in Older Adults: The HAR70+ Model. Sensors, 23(5), 2368. 

2. The research has not concluded the optimal time window for the HAR in their conclusion section and results discussion section.

Response: 

Thank you for pointing this out. 

In our results discussion section, we acknowledge the importance of the time window size, and we conclude that it is dependent on the target activity. Specifically, we highlight that the two 1-second models outperform others across test groups and most activities, except for cycling, where 3- or 5-second windows show better performance. Page 18-19, lines 332-335 and 337-340. We would also like to refer to the section where we discuss the window settings and the preference of different time windows on page 21-22, lines 403-440.

We have revised our conclusion to provide a more precise summary of our findings. On page 25, lines 499-507 now state:

"Based on the highest overall accuracy, the NTNU HAR-ChildrenCP model with 1-second window would be recommended for all three test groups. However, the optimal window size and overlap depend on the target activity. For activities such as cycling, 3- or 5-second windows perform better. Considering the ability to predict both momentary activities and activities of extended duration, the 3-second window without overlap would be recommended for population measurements."

Reviewer #3:

The authors provided convincing answers to my concerns and consequently modified the article. There is only one concern for which extra information would be useful. Regarding robustness to sensors' misplacement, the authors explained that "Our research group have tested the consequence of sensor misplacement on model performance in previous studies using

the same methodology (unpublished). As long as the sensors are placed in the correct direction towards gravity minor deviations in placements does not affect the accuracy of the model." Even if unpublished yet, the results

---

## [Decision Letter · Decision Letter 2]

24 Jun 2024

PONE-D-23-37915R2Validation of two novel human activity recognition models for typically developing children and children with Cerebral Palsy.PLOS ONE

Dear Dr. Tørring,

Thank you for submitting your manuscript to PLOS ONE. After careful consideration, we feel that it has merit but does not fully meet PLOS ONE’s publication criteria as it currently stands. Therefore, we invite you to submit a revised version of the manuscript that addresses the points raised during the review process.

Despite the reviewers agreed about the significant revisions made by the authors, they are still not convinced on the novelty and technical contribution of the work. The authors should present the model file and include some results for deep learning-based experiments in the manuscript to improve the technical contributions of the work adequately. 

We look forward to receiving your revised manuscript.

Kind regards,

Jyotindra Narayan

Academic Editor

PLOS ONE

Reviewers' comments:

Reviewer's Responses to Questions

**Comments to the Author**

1. If the authors have adequately addressed your comments raised in a previous round of review and you feel that this manuscript is now acceptable for publication, you may indicate that here to bypass the “Comments to the Author” section, enter your conflict of interest statement in the “Confidential to Editor” section, and submit your "Accept" recommendation.

Reviewer #2: All comments have been addressed

Reviewer #3: All comments have been addressed

2. Is the manuscript technically sound, and do the data support the conclusions?

Reviewer #2: Partly

Reviewer #3: (No Response)

3. Has the statistical analysis been performed appropriately and rigorously? 

Reviewer #2: Yes

Reviewer #3: (No Response)

4. Have the authors made all data underlying the findings in their manuscript fully available?

Reviewer #2: No

Reviewer #3: (No Response)

5. Is the manuscript presented in an intelligible fashion and written in standard English?

Reviewer #2: Yes

Reviewer #3: (No Response)

6. Review Comments to the Author

Reviewer #2: The manuscript lacks significant technical contribution except for noble datasets. However, there had been few publicly available HAR datasets ( but not with children and cerebral palsy). Several works have already been on HAR for different age groups and people with cerebral palsy. However, as per the author's claim, this is the first work on HAR in children with Cerebral Palsy.

The GitHub repository https://github.com/ntnu-ai-lab/harth-ml-experiments, provided by the user, contains well-documented code, except the saved model files. The author should publish the model file in the GitHub repo for the reproducibility of the result. The GitHub repository already contains the code for BILSTM and CNN on the same datasets. Therefore, the authors are requested to include the results of the deep-learning-based experiments (biLSTM, CNN) in the current manuscript to justify the technical contribution and the PLOS ONE journal's reputation. There are already a few existing resources (research papers, HAR datasets, GitHub public repository containing ML, Deep learning and other methods for HAR). There has been no significant improvement in technical contribution from the first submission except for the improvement in writing and clarity of the manuscript.

Some references (published articles, HAR datasets, public GitHub repositories) are provided below:

Existing related papers on the HAR:

1. D. Ravi, C. Wong, B. Lo and G. -Z. Yang, "Deep learning for human activity recognition: A resource efficient implementation on low-power devices," 2016 IEEE 13th International Conference on Wearable and Implantable Body Sensor Networks (BSN), San Francisco, CA, USA, 2016, pp. 71-76, doi: 10.1109/BSN.2016.7516235. keywords: {Machine learning;Feature extraction;Spectrogram;Convolution;Time-frequency analysis;Data mining;Deep Learning;Low-Power Devices;HAR;ActiveMiles},

2. M. Mostafavizadeh, A. R. Sadri and M. Zekri, "Walking pattern classification in children with cerebral palsy: A wavelet network approach," The 16th CSI International Symposium on Artificial Intelligence and Signal Processing (AISP 2012), Shiraz, Iran, 2012, pp. 243-249, doi: 10.1109/AISP.2012.6313752. keywords: {Classification algorithms;Acceleration;Force;Legged locomotion;Entropy;Feature extraction;Accelerometers;Cerebral Palsy;kinetic data;Accelerometer;Pattern Classification;Wavelet Network;Shannon entropy},

3. Csizmadia, G., Liszkai-Peres, K., Ferdinandy, B. et al. Human activity recognition of children with wearable devices using LightGBM machine learning. Sci Rep 12, 5472 (2022). https://doi.org/10.1038/s41598-022-09521-1

4. Taborri, J.; Scalona, E.; Palermo, E.; Rossi, S.; Cappa, P. Validation of Inter-Subject Training for Hidden Markov Models Applied to Gait Phase Detection in Children with Cerebral Palsy. Sensors 2015, 15, 24514-24529. https://doi.org/10.3390/s150924514

5. J. Kamruzzaman and R. K. Begg, "Support Vector Machines and Other Pattern Recognition Approaches to the Diagnosis of Cerebral Palsy Gait," in IEEE Transactions on Biomedical Engineering, vol. 53, no. 12, pp. 2479-2490, Dec. 2006, doi: 10.1109/TBME.2006.883697.

6. Pengxi Fu, Jianxin Guo, Hongxiang Luo, LightGBM for Human Activity Recognition Using Wearable Sensors. Automation and Machine Learning (2024) Vol. 5: 113-118. DOI: http://dx.doi.org/10.23977/autml.2024.050114

7. Malekzadeh, M., Clegg, R., Cavallaro, A., & Haddadi, H. (2021). Dana: Dimension-adaptive neural architecture for multivariate sensor data. Proceedings of the ACM on Interactive, Mobile, Wearable and Ubiquitous Technologies, 5(3), 1-27.

8. Phyo, C. N., Zin, T. T., & Tin, P. (2019). Deep learning for recognizing human activities using motions of skeletal joints. IEEE Transactions on Consumer Electronics, 65(2), 243-252.

HAR datasets:

1. UCI-HAR : https://archive.ics.uci.edu/dataset/344/heterogeneity+activity+recognition - containing 30 users performing 6 activities. Accelerometer and gyroscope data were collected by a smartphone worn on the waist

2. UTwente: https://www.mdpi.com/1424-8220/16/4/426 includes data of 10 users performing 13 activities using accelerometer, gyroscope, and magnetometer data collected from the device on the (right) wrist.

3. MobiAct: https://www.scitepress.org/papers/2016/57924/57924.pdf

4. MotionSense: https://arxiv.org/abs/1802.07802

Public Github repository on HAR:

1. https://github.com/mmalekzadeh/motion-sense/tree/master

2. https://github.com/mmalekzadeh/dana

3. https://github.com/guillaume-chevalier/LSTM-Human-Activity-Recognition

4. https://github.com/aqibsaeed/Human-Activity-Recognition-using-CNN

Reviewer #3: (No Response)

7. PLOS authors have the option to publish the peer review history of their article (what does this mean?). If published, this will include your full peer review and any attached files.

Reviewer #2: **Yes: **Dr. Durgesh Kumar

Reviewer #3: **Yes: **Armel Crétual

---

## [Author Response · Author response to Decision Letter 2]

26 Jul 2024

Response to editor and reviewer 

Our response is provided under the comments, page and line number refer to the manuscript file with track changes. 

Dear Dr. Tørring,

Thank you for submitting your manuscript to PLOS ONE. After careful consideration, we feel that it has merit but does not fully meet PLOS ONE’s publication criteria as it currently stands. Therefore, we invite you to submit a revised version of the manuscript that addresses the points raised during the review process.

Despite the reviewers agreed about the significant revisions made by the authors, they are still not convinced on the novelty and technical contribution of the work. The authors should present the model file and include some results for deep learning-based experiments in the manuscript to improve the technical contributions of the work adequately. 

We look forward to receiving your revised manuscript.

Kind regards,

Jyotindra Narayan

Academic Editor

PLOS ONE

Response to editor: 

Dear editor, Jyotindra Narayan 

Thank you for your understanding and for your considerate suggestion. We appreciate the time and effort you have dedicated to evaluating our manuscript and guidance in the current situation. We would like to refer to our email correspondence for further information about our position. 

While we have decided not to engage in an extensive round of revisions that would alter the original scope of our study, we will proceed with making the minimal revisions as recommended. 

Specifically, we will:

1. Clarify the novelty of our work, once again emphasizing the clinical relevance of our study focused on children with CP.

2. Provide the model file trained on all subjects to enhance reproducibility on GitHub. 

We have also made changes to our manuscript to highlight these points. 

Finally, we would once again emphasize that two of the reviewers have already accepted our manuscript. Therefore, we ask you to make an editorial decision based on all three reviewers’ remarks. Thank you once again for your guidance and support. We look forward to your final decision.

Best regards,

Marte Fossflaten Tørring

6. Review Comments to the Author

Reviewer #2: The manuscript lacks significant technical contribution except for noble datasets. However, there had been few publicly available HAR datasets ( but not with children and cerebral palsy). Several works have already been on HAR for different age groups and people with cerebral palsy. However, as per the author's claim, this is the first work on HAR in children with Cerebral Palsy.

Response: 

We appreciate your recognition of the unique dataset we have presented.

As we mentioned in our previous response, we see our novelty not in technical innovation but in the utilization of a technical methods in real-world scenarios and clinical settings. We would like to clarify a few points regarding the novelty and technical contribution of our work:

1. Not the First HAR Model for Children or Children with CP: While it is true that there have been previous HAR models for children and individuals with Cerebral Palsy, our work differentiates itself by achieving a level of accuracy that makes it practically employable in clinical studies. Our model demonstrates a significant improvement in performance, making it a reliable tool for clinical applications.

2. Addressing Childlike Behavior and Free Play: Our study is the first to specifically address the nuances of childlike behavior and free play in children with Cerebral Palsy. This aspect is critical, as it aligns the HAR model more closely with real-world scenarios and clinical needs. The listed machine learning models and HAR experiments cited by the reviewer do not appear to incorporate this important dimension, which is essential for developing clinically relevant tools.

We believe that these contributions significantly enhance the technical value of our work and its potential impact in the field of activity monitoring for children with Cerebral Palsy.

In our introduction on page 4, lines 79-86, we have presented existing HAR models for children with CP and explained why our HAR model is still needed. This is further emphasized in our aim on page 5, lines 99-101.

Reviewer #2:

The GitHub repository https://github.com/ntnu-ai-lab/harth-ml-experiments, provided by the user, contains well-documented code, except the saved model files. The author should publish the model file in the GitHub repo for the reproducibility of the result. The GitHub repository already contains the code for BILSTM and CNN on the same datasets. Therefore, the authors are requested to include the results of the deep-learning-based experiments (biLSTM, CNN) in the current manuscript to justify the technical contribution and the PLOS ONE journal's reputation. There are already a few existing resources (research papers, HAR datasets, GitHub public repository containing ML, Deep learning and other methods for HAR). There has been no significant improvement in technical contribution from the first submission except for the improvement in writing and clarity of the manuscript.

Response: 

We appreciate your suggestion to enhance the reproducibility of our results. To this end, we will publish the model file in our GitHub repository (https://github.com/ntnu-ai-lab/harth-ml-experiments). We have made changes to our manuscript on page 10, line 211. 

Regarding the additional deep-learning-based experiments, we would like to refer to our previous response in not altering our scope and not going beyond our aim of a clinically relevant tool. Thereby we would like to clarify a few points:

1. GitHub Repository Content: The existing code in our GitHub repository does not include implementations of biLSTM and CNN models for the specific dataset used in our current study. The code available is for different datasets, and applying these models to our dataset would require extensive additional work, including data preprocessing, hyperparameter tuning, and prolonged training periods.

2. Scope of Our Study: As we have emphasized in previous responses, our study's primary goal is to address a specific clinical need using established methods. Expanding the experiments to include deep learning models like biLSTM and CNN would significantly alter the scope of our research and deviate from our original objectives. Our focus has been on creating a practical, clinically relevant tool, rather than exploring a wide array of technical methodologies.

3. Technical Contribution: We believe that our contributions lie in the successful application of technical methods to real-world clinical settings, achieving a level of accuracy that makes our model practically employable. Additionally, our work addresses childlike behavior and free play in children with Cerebral Palsy, a critical aspect that has not been incorporated in existing models.

We hope this clarifies our position and the rationale behind our decisions not to do additional experiments. We are committed to making our work as reproducible and transparent as possible.

---

## [Editor Report · Decision Letter 3]

1 Aug 2024

Validation of two novel human activity recognition models for typically developing children and children with Cerebral Palsy.

PONE-D-23-37915R3

Dear Dr. Tørring,

We’re pleased to inform you that your manuscript has been judged scientifically suitable for publication and will be formally accepted for publication once it meets all outstanding technical requirements.

Kind regards,

Jyotindra Narayan

Academic Editor

PLOS ONE

Additional Editor Comments (optional):

Following the successive revisions and authors' reponse to the reviewer concerns, the mansucript is now recommened for acceptance and publication. Congratulations to the authors fo the good work.
---

## [Editor Report · Acceptance letter]

11 Sep 2024

PONE-D-23-37915R3 

PLOS ONE

Dear Dr. Tørring, 

I'm pleased to inform you that your manuscript has been deemed suitable for publication in PLOS ONE. Congratulations! Your manuscript is now being handed over to our production team.

Kind regards, 

on behalf of

Dr. Jyotindra Narayan 

Academic Editor

PLOS ONE